# Specific ablation of PDGFRβ-overexpressing pericytes with antibody-drug conjugate potently inhibits pathologic ocular neovascularization in mouse models

Seok Jae Lee [1,2,12], Soohyun Kim [3,4,5,10,11,12], Dong Hyun Jo[6], Chang Sik Cho[1], Su Ree Kim[7], Dongmin Kang[7], Jisu Chae[3,4], Duck Kyun Yoo[2,3,5], Suji Ha[3,4], Junho Chung [2,3,4,5 ✉] & Jeong Hun Kim[1,2,8,9 ✉]

## Abstract

**Background** Crosstalk between pericytes and endothelial cells is critical for ocular neovascularization. Endothelial cells secrete platelet-derived growth factor (PDGF)-BB and recruit PDGF receptor β (PDGFRβ)–overexpressing pericytes, which in turn cover and stabilize neovessels, independent of vascular endothelial growth factor (VEGF). Therapeutic agents inhibiting PDGF-BB/PDGFRβ signaling were tested in clinical trials but failed to provide additional benefits over anti-VEGF agents. We tested whether an antibody-drug conjugate (ADC) – an engineered monoclonal antibody linked to a cytotoxic agent - could selectively ablate pericytes and suppress retinal and choroidal neovascularization.

**Methods** Immunoblotting, flow cytometry, cell viability test, and confocal microscopy were conducted to assess the internalization and cytotoxic effect of ADC targeting mPDGFRβ in an in vitro setting. Immunofluorescence staining of whole-mount retinas and retinal pigment epithelium-choroid-scleral complexes, electroretinography, and OptoMotry test were used to evaluate the effect and safety of ADC targeting mPDGFRβ in the mouse models of pathologic ocular neovascularization.

**Results** ADC targeting mPDGFRβ is effectively internalized into mouse brain vascular pericytes and showed significant cytotoxicity compared with the control ADC. We also show that specific ablation of PDGFRβ-overexpressing pericytes using an ADC potently inhibits pathologic ocular neovascularization in mouse models of oxygen-induced retinopathy and laser-induced choroidal neovascularization, while not provoking generalized retinal toxicity.

**Conclusion** Our results suggest that removing PDGFRβ-expressing pericytes by an ADC targeting PDGFRβ could be a potential therapeutic strategy for pathologic ocular neovascularization.

**Plain language summary**

Many diseases of the eye, such as age-related macular degeneration, involve abnormal blood vessel formation in the eye. One way to treat these diseases is to block the formation of blood vessels, for example by targeting cells called pericytes that surround small blood vessels. Here, we test a pericyte-targeting antibody-drug conjugate – in which a protein that can target a specific molecule on the cell surface is chemically linked to a drug that kills cells – for its ability to block blood vessel formation. We demonstrate that this approach is safe and effective in two mouse models. Our findings suggest that this might be a potentially useful strategy to treat these diseases in patients, for example when existing therapies no longer work.

[1] Fight against Angiogenesis-Related Blindness (FARB) Laboratory, Clinical Research Institute, Seoul National University Hospital, Seoul, Republic of Korea. [2] Department of Biomedical Sciences, Seoul National University College of Medicine, Seoul, Republic of Korea. [3] Department of Biochemistry and Molecular Biology, Seoul National University College of Medicine, Seoul, Republic of Korea. [4] Cancer Research Institute, Seoul National University College of Medicine, Seoul, Republic of Korea. [5] Transplantation Research Institute, Seoul National University College of Medicine, Seoul, Republic of Korea. [6] Department of Anatomy & Cell Biology, Seoul National University College of Medicine, Seoul, Republic of Korea. [7] Department of Life Science, Fluorescence Core Imaging Center, Ewha Womans University, Seoul, Republic of Korea. [8] Department of Ophthalmology, Seoul National University College of Medicine, Seoul, Republic of Korea. [9] Advanced Biomedical Research Center, Korea Research Institute of Bioscience & Biotechnology, Daejeon, Republic of Korea. [10]Present address: Department of Biochemistry, Stanford University School of Medicine, Stanford, CA 94305, USA. [11]Present address: Stanford ChEM-H, Stanford University, Stanford, CA 94305, USA. [12]These authors contributed equally: Seok Jae Lee, Soohyun Kim. ✉email: jjhchung@snu.ac.kr; steph25@snu.ac.kr

Monoclonal antibody-based treatments have been employed for the treatment of diverse ocular neovascular diseases, including wet-type age-related macular degeneration (AMD), polypoidal choroidal vasculopathy, myopic choroidal neovascularization (NV), macular edema secondary to diabetic retinopathy, and retinal vascular occlusion[1–4]. Notably, intravitreal anti-vascular endothelial growth factor (VEGF) therapy is widely used to treat ocular NV because intraocular VEGF secreted by vascular/extravascular components is one of the strongest inducers of ocular angiogenesis[5–9]. Although VEGF-specific agents are the standard treatment for wet-type AMD[2,10,11], ~20% of patients are nonresponders and ultimately lose their visual acuity[12–16]. In addition, repeated intravitreal injections of anti-VEGF agents can lead to several adverse events, including vitreoretinal fibrosis, chorioretinal atrophy, and sub-retinal/sub-retinal pigment epithelium (RPE) fluid or hemorrhage[17,18]. Thus, more effective therapeutic agents are required for ocular NV, either as stand-alone monotherapy or in combination with anti-VEGF agents.

In addition to VEGF, several other molecules participate in the development of retinal and choroidal NV, including platelet-derived growth factor (PDGF), placental growth factor, erythropoietin, stromal-derived factor-1, and angiopoietins. Although numerous basic studies and clinical trials utilizing new monoclonal antibodies that target various angiogenesis-related molecules and signaling pathways have been conducted[19–21], most of these antibodies were not as efficacious in combination therapies as anti-VEGF monotherapy, when studied in clinical settings. In particular, PDGF-BB/PDGF receptor β (PDGFRβ) signaling inhibitors were regarded as promising additive to conventional anti-VEGF therapy. PDGF-BB is the predominant isoform of PDGF in the ocular system, which is primarily expressed by vascular endothelial cells[22–24], and PDGFRβ is commonly expressed on pericytes, smooth muscle cells, vascular endothelial cells, and RPE[25–27]. The ligand/receptor pair of PDGF-BB/PDGFRβ induces the recruitment of pericytes to neovessels, which then cover these vessels[28]. This pair also induces the secretion of VEGF and other cell survival factors from pericytes[23,29,30]. Subsequently, these protective functions of pericytes enable endothelial cell survival independent of VEGF, rendering anti-VEGF therapy less efficient[31–34]. In the laser-induced choroid NV (CNV) model, laser injury promotes the migration of pericyte-like cells into the area of CNV lesions and induces their proliferation, leading to the expression of markers such as smooth muscle actin and PDGFRβ[35,36]. These pericyte-like cells form a scaffold for subsequent vascular endothelial cell infiltration and NV[37]. PDGF-BB/PDGFRβ signaling inhibitors have demonstrated additive benefits for suppressing CNV in vivo when employed in combination with anti-VEGF agents[37,38]. However, in clinical settings, PDGF/PDGFRβ signaling inhibitors have not provided additive benefits in combination with anti-VEGF agents, when compared with anti-VEGF monotherapy, and have failed to improve the pathologic anatomy of the retina and choroid or enhance visual acuity[39].

We hypothesized that therapy combining PDGF/PDGFR signaling inhibitors with anti-VEGF agents is insufficient to effectively ablate pericytes and inhibit the PDGF/PDGFR-related signaling pathway in the pathologic ocular environment. Studies have reported an inverse correlation between concentrations of PDGF/PDGFRβ and VEGF during anti-VEGF treatments[40–42], as well as excessive coverage of PDGFRβ-positive cells in CNV lesions of patients resistant to anti-VEGF therapy[43]. In the current study, we therefore tested whether specific delivery of cytotoxic reagents to PDGFRβ-overexpressing cells are effective in reducing pathologic ocular NV.

Antibody-drug conjugates (ADCs) are monoclonal antibodies linked to cytotoxic effector molecules that induce cell death upon binding and internalization of the antibody by the target cell. To date, nine ADCs targeting human epidermal growth factor receptor-2 (HER2), CD22, CD30, CD33, CD79B, Nectin-4, and Trop-2 are clinically available for the treatment of HER2-positive metastatic breast cancer, B-cell acute lymphoblastic leukemia, Hodgkin's lymphoma, acute myeloid leukemia, B-cell lymphoma, bladder cancer, and triple-negative breast cancer, respectively[44]. Furthermore, over 100 ADCs targeting various cancers are under evaluation in clinical trials[45]. However, until now, ADCs have rarely been evaluated outside oncologic applications, although a few studies have reported such uses of ADCs[46,47]. In a series of studies, we developed a unique ADC composed of hapten-conjugated drugs and bispecific single-chain variable fragment (scFv)- human kappa constant ($C_\kappa$)-scFv fusion protein that can bind simultaneously to hapten and antigen of interest. The advantages of this platform are numerous. Cotinine, a major metabolite of nicotine, was used as a hapten due to its non-toxicity, physiological inertness, and exogeneity[48]. With commercial availability of trans-4-cotinine carboxylic acid, cotinine can be chemically cross-linked to various cytotoxic drugs and linkers with high purity[49–51]. Furthermore, as the drug does not require direct conjugation to the antibody, it does not alter its stability and affinity. Another advantage of this platform is that the ADC can be formed by simply mixing cotinine-drug conjugate and bispecific antibody. We have previously demonstrated that bispecific anti-mPDGFRβ × cotinine scFv-$C_\kappa$-scFv fusion protein (anti-mPDGFRβ × cotinine) and cotinine-conjugated duocarmycin could specifically induce cytotoxicity toward PDGFRβ-expressing cells[51]. In the current study, we tested whether this ADC could ablate pericytes and suppress retinal and choroidal NV in oxygen-induced retinopathy (OIR) and laser-induced CNV (LI-CNV) mouse models. ADC targeting mPDGFRβ effectively removed PDGFRβ-expressing pericytes and led to the amelioration of pathological NV in the mouse models, with no visible signs of toxicity.

## Materials and methods

**Cell culture.** Mouse brain vascular pericytes (MBVPs) were obtained from iXCells Biotechnologies (San Diego, CA, USA) and maintained in mouse pericyte growth medium (MD-0092; iXCells Biotechnologies) supplemented with 1% penicillin/streptomycin (Gibco, Grand Island, NY, USA).

**Preparation of bispecific scFv-$C_\kappa$-scFv fusion protein and ADC targeting mPDGFRβ.** Anti-mPDGFRβ × cotinine was expressed and purified as described previously[51]. Clone PRb-CN01 developed in our previous study was used for anti-mPDGFRβ scFv. As a control, we also prepared anti-HER2 × cotinine bispecific scFv-$C_\kappa$- scFv fusion protein (anti-HER2 × cotinine), where anti-HER scFv was derived from trastuzumab[51]. The expression vectors encoding these bispecific scFv-$C_\kappa$-scFv fusion proteins were transfected into Expi293F cells (Thermo Fisher Scientific Inc., Waltham, MA, USA), and the scFv-$C_\kappa$-scFv fusion proteins were purified using KappaSelect resin (GE Healthcare, Buckinghamshire, UK) following the manufacturer's instructions.

ADC targeting mPDGFRβ was prepared by mixing anti-mPDGFRβ × cotinine (1 μM) with cotinine-duocarmycin (1 μM) at a 1:1 molar ratio as described previously[51]. The complex was then incubated for 30 min at room temperature to allow complex formation before being used. Control ADC was prepared in the same way. The schematic diagrams of a bispecific antibody, cotinine-duocarmycin, and the conjugation process are represented

in Supplementary Fig. 6. The LC-MS analysis and RP-HPLC analysis that empirically determine the drug-to-antibody ratio were performed by Levena Biopharma (San Diego, CA, USA) and the data are shown in Supplementary Fig. 7.

**Flow cytometry**. MBVPs were incubated with either anti-mPDGFRβ × cotinine (100 nM) or control scFv-Cκ-scFv fusion protein in flow cytometry buffer (1% [w/v] BSA in Phosphate-buffered saline (PBS) containing 0.05% [w/v] sodium azide) at 4 °C for 1 h. After washing four times with the flow cytometry buffer, the cells were probed with APC-conjugated anti-human Cκ antibody (1:100 dilution, clone TB28-2; BD Biosciences, San Jose, CA, USA). For each sample, the data were acquired from 10,000 cells, and the results were analyzed using FlowJo (Tree Star, Ashland, OR, USA).

**Confocal microscopy**. MBVPs were incubated with mouse pericyte growth medium containing anti-mPDGFRβ × cotinine (10 µg/mL) with or without mPDGF-BB (50 ng/mL; 315-18; Peprotech, Rocky Hill, NJ, USA) for 30 min at 37 °C as described previously[51–53]. Next, the cells were blocked by incubation with 0.1% Triton X-100 and 5% horse serum (GIBCO) in PBS for 30 min. Subsequently, the cells were incubated with FITC-conjugated anti-human Cκ antibody (2 µg/mL; TB28-2; BD Biosciences) for 30 min at room temperature. Early endosomes were imaged by incubating the cells with 1:200 diluted anti-Rab5 antibody (#C8B1; Cell Signaling Technology, Danvers, MA, USA) and Alexa Fluor 546-conjugated goat anti-rabbit IgG (#A-11035; Invitrogen, Carlsbad, CA, USA). To detect cellular DNA, 4′,6-diamidino-2-phenylindole (DAPI, 0.2 µg/mL) was used as described previously[53]. Confocal images were acquired using a Zeiss LSM 880 microscope at Ewha Fluorescence Core Imaging Center, and the images were processed with Zen software (Carl Zeiss, Thornwood, NY, USA).

**Cytotoxicity assay**. Cytotoxicity assays were performed as described previously[51,53]. Briefly, 4000 MBVP cells in 50 µL mouse pericyte growth medium were seeded in 96-well plates (#CLS3595; Corning Inc., Corning, NY, USA) and incubated overnight at 37 °C in a humidified atmosphere with 5% CO$_2$. ADCs (2 µM) were diluted by 5 folds (0.024 nM to 2 µM) and mixed 1:1 with medium with or without mPDGF-BB (8 nM). ADCs [0.012 nM to 1 µM] with or without mPDGF-BB (4 nM), at a total volume of 50 µL, were added to the pre-seeded cells in 50 µL of medium, yielding a total volume of 100 µL. ADCs [0.006–500 nM] with or without mPDGF-BB (2 nM) were then incubated with the cells for 72 h at 37 °C in a humidified atmosphere with 5% CO$_2$.

**Mice**. C57BL/6J mice were maintained in a specific pathogen-free facility at Seoul National University. The mice (aging-matched: 6-week-old, weight range: 20–24 g) were allocated into experimental groups. The total number of mice used in each experiment was determined based on the preliminary result of the OIR and LI-CNV mouse study using ADC. No statistical methods were used to predetermine sample size. All animal procedures were approved by the Seoul National University Animal Care and Use Committee (Permit Number: SNU-171203-1-2) and were conducted following the guidelines of the Association for Research in Vision and Ophthalmology Statement for the use of Animals in Ophthalmic and Vision Research.

**Preparation of whole-mount retinas and RPE-choroid-scleral complexes from normal mice and immunofluorescence staining**. The eyes from 6-week-old wild-type C57BL/6J male mice

were enucleated and fixed with 4% paraformaldehyde (#P2031; Biosesang, Seongnam, Gyeonggi-do, Korea) for 15 min at room temperature. After washing with PBS, the eyes were gently dissected to remove all components except the retina and RPE-choroid-scleral (RCS) complexes. The retina and RCS complexes were whole mounted and incubated in Perm/Block solution (0.2% Triton X-100 and 0.3% BSA in PBS) at room temperature for 1 h. Next, the samples were incubated overnight at 4 °C with rabbit anti-PDGFRβ antibody (1:100; #3169, Cell Signaling Technology, Danvers, MA, USA). After washing with PBS, the samples were incubated at room temperature for 2 h with Alexa Fluor 647-labeled donkey anti-rabbit IgG antibody (1:400; #A31573; Invitrogen, Carlsbad, CA, USA). After washing with PBS, the samples were stained with Alexa Fluor 568-conjugated anti-IB4 antibody (1:400; #I21412; Invitrogen, Carlsbad, CA, USA) and Alexa Fluor 488-conjugated anti-NG2 antibody (1:400; #AB5320A4; Sigma-Aldrich, St. Louis, MO, USA) at room temperature for 2 h. After washing with PBS, the samples were counterstained with 10 mg/ml of DAPI (1:1000; #D9542; Sigma-Aldrich, St. Louis, MO, USA) at room temperature for 15 min. After washing with PBS, the retina and RCS complexes were mounted with Fluoromount™ Aqueous Mounting Medium (#F4680, Sigma-Aldrich, St. Quentin, France) and observed under a confocal microscope (Leica TCS STED; Leica Microsystems Ltd., Wetzlar, Germany).

**Induction of oxygen-induced retinopathy**. OIR was induced in C57BL/6J wild-type mice ($n = 6$ per group, $n = 18$ in total). Briefly, newborn pups (male and female) and their nursing dam were placed in 75 ± 0.5% oxygen in an O$_2$-regulated chamber with an oxygen controller (Pro-Ox 110 Chamber Controller; Biospherix, Redfield, NY, USA) from P7 to P12 and then returned to room air. Next, the mice were divided into three groups (six mice per group). On P14, each group received an intravitreal injection of PBS as a vehicle control, control ADC (66.77 pg), ADC targeting mPDGFRβ (66.77 pg). To administer the indicated reagents intravitreally, a microliter syringe with a 33G blunt needle (Hamilton Bonaduz AG, Bonaduz, Switzerland) was inserted into the vitreous cavity through the 6 o'clock position of the limbus with a 45° injection angle for the right eye, followed by gently loading the solutions under the surgical microscope (Leica Microsystems, Ltd., Wetzlar, Germany). Intravitreal injection was only conducted in the right eye. After euthanization of the mice on P17, the retina was isolated, fixed, and mounted. After immunofluorescence staining with Alexa Fluor 568-conjugated anti-IB4 antibody (1:200; #I21412; Invitrogen, Carlsbad, CA, USA) and Alexa Fluor 488 conjugated anti-NG2 antibody (1:200; #AB5320A4; Sigma-Aldrich, St. Louis, MO, USA) as described previously, manifestations of oxygen-induced vascular pathology (avascular area and neovascular tufts) were visualized and imaged using a confocal microscope. Area quantification was performed using ImageJ 1.42 software (National Institutes of Health, Bethesda, MD, USA). For analyzing NG2 or PDGFRβ coverage to neovascular tufts and peripheral retinal vasculature, we randomly choose a sampling box of four sites per eye at the mid-periphery region (neovascular tufts area) and peripheral region (peripheral vascularized area), respectively in the whole-mounted retinas and converted to 3D immunofluorescence image format using a built-in measuring tool of the LAS X system (Leica Microsystems Ltd., Wetzlar, Germany).

**Laser-induced choroidal neovascularization**. Six-week-old wild-type C57BL/6J male mice ($n = 6$ per group, $n = 18$ in total) were fully anesthetized by intraperitoneal injection using a mixture (3:1 ratio, 1 mL/kg) of zolazepam and tiletamine (Zoletil 50®, Virbac, Carros, France) and xylazine (Rompun®, Bayer Korea, Seoul,

Korea). After dilating the pupils with tropicamide 1% (Tropherin®, Hanmi. Pharm Co. Ltd., Seoul, Korea), a laser photocoagulator with an indirect headset delivery system (Ilooda, Suwon, Gyeonggi-do, Korea) was used to visualize the retina. The laser parameters were as follows: wavelength: 810 nm; spot size: 200 μm; power: 1 W; and exposure time: 100 msec. Sufficient laser energy was delivered to four locations for each eye (the 3, 6, 9, and 12 o'clock positions of the posterior pole around the optic disc) to induce rupture of Bruch's membrane. Only burns that produced a bubble without vitreous hemorrhage were included in the study. The mice were then divided into three groups (six mice per group). On day 7 after laser photocoagulation, the six mice in each group received an intravitreal injection of PBS as a vehicle control, control ADC (667.7 pg), and ADC targeting mPDGFRβ (667.7 pg). To evaluate the effect of ADC on choroidal neovascularization, the eyes were enucleated, fixed in 4% paraformaldehyde, and then prepared for whole-mounted RCS complex formation. After immunofluorescence staining described above, CNV area and volume, NG2 volume, and PDGFRβ volume of all the laser burn sites (four laser burn sites for each eye) were quantitatively analyzed using a built-in measuring tool, the LAS X systems (Leica Microsystems Ltd., Wetzlar, Germany).

**Histologic evaluation of retinas from mice treated with ADC targeting mPDGFRβ**. ADC targeting mPDGFRβ (667.7 pg), control ADC (667.7 pg), or PBS was injected intravitreally into 6-week-old wild-type C57BL/6J male mice ($n = 6$ per group, $n = 18$ in total). One week after the injection, the eyes were enucleated, fixed in 4% paraformaldehyde, and embedded in paraffin. After 4-μm-thick paraffin sections were prepared, the sections were deparaffinized and hydrated by sequential immersion in xylene substitutes and graded ethyl alcohol solutions. Next, the sections were processed by either hematoxylin and eosin (H&E) or terminal deoxynucleotidyl transferase dUTP nick end labeling (TUNEL) staining. To evaluate changes in the retinal layers, the retinal layer thickness ratio was calculated as follows: retinal thickness from the internal limiting membrane to the inner nuclear layer/retinal thickness from the internal limiting membrane to the outer nuclear layer. This was compared with the ratio in control mice. TUNEL staining was performed using an in situ cell death detection kit (#1164795910; Sigma-Aldrich, St Louis, MO, USA). The nuclei were counterstained with DAPI, and TUNEL-positive cells were counted in five randomly selected fields in each slide (×200) under a fluorescence microscope.

**Electroretinography**. To assess the retinal function by electroretinography (ERG), the mice were dark-adapted for over 16 h before electroretinogram monitoring. After deep anesthesia and dilatation of the pupils with an eye drop containing phenylephrine hydrochloride (5 mg/ml) and tropicamide (5 mg/ml), the mice were placed on a heating pad in a Ganzfeld dome to maintain their body temperature. The contact lens electrode was placed in the center of the cornea after instilling artificial tears. The reference electrode was located at the forehead, and the ground electrode was located at the tail. Full-field ERG was recorded using the universal testing and electrophysiologic system (UTAS E-3000; LKC Technologies Inc., Gaithersburg, MD, USA). In the dark-adapted condition, the scotopic responses were recorded using a series of white flashes of increasing intensities ranging from −5.0 to 0.0 log (cd·s/m$^2$) using a notch filter at 60 Hz and a digital bandpass filter ranging from 0.3 to 500 Hz. After the completion of the dark-adapted stimulus series, the mice were projected with a steady field of light (20 cd·s/m$^2$) for 15 min to desensitize rods. In the light-adapted condition with 1.3 log (cd·s/m$^2$) background, the photopic responses were

recorded ranging from 0.39 to 1.39 log (cd·s/m$^2$) with the filter ranging from 2 to 200 Hz. After averaging the signals, the amplitudes were measured by the built-in software (UTAS visual electrodiagnostic system with EMWin, LKC Technologies Inc., Gaithersburg, MD, USA) and used for analysis.

**OptoMotry test**. A virtual optomotor system (OptoMotry apparatus; CerebralMechanics Inc., Lethbridge, Alberta, Canada) was used to assess visual function. Briefly, the mice were placed on an elevated platform positioned in the middle of an arena created by four inward-facing display monitors. Spatial frequency thresholds were assessed using a video camera to monitor the elicitation of the optokinetic reflex through virtual stimuli projected with sine-wave gratings (100% contrast) on the computer monitors. Experimenters were blinded to the treatment and each animal's previously recorded thresholds.

**Statistics and reproducibility**. Statistical analyses were performed using SPSS software version 22.0 (SPSS Inc., Chicago, IL, USA) and GraphPad Prism v.8.0.1. The experimental data were presented as mean ± standard error of the mean. $P$ values were determined using one-way ANOVA and Tukey post-hoc tests for multiple groups. All the data in our manuscript were repeated at least three times independently with similar results. The experiments were not randomized and the investigators were not blinded to allocation during experiments and outcome assessment.

## Results

**PDGFRβ is highly expressed on pericytes in pathologic neovessels of OIR and LI-CNV**. To monitor the expression patterns of PDGFRβ in the retina and RPE of OIR and LI-CNV mice, eyeballs were retrieved after inducing retinal and choroidal neovascularization. Next, the whole retina and RPE were mounted and stained with anti-PDGFRβ, anti-isolectin B4 (IB4), or anti-neural/glial antigen 2 (NG2) antibodies. We also stained the corresponding tissues of wild-type mice and found that PDGFRβ was present mainly in the retinal perivascular area and surrounding tissues, from the superficial vascular plexus to the deep vascular plexus (Fig. 1a). Furthermore, in OIR mice on postnatal day 17, PDGFRβ was highly overexpressed around pathologic vessels, particularly around the neovascular tufts (Fig. 1b). The CNV lesion also showed dramatically increased levels of PDGFRβ, compared with control RPE (Fig. 1c, d). Co-localization of the vascular endothelial cell marker IB4 and PDGFRβ was negligible; however, the pericyte marker NG2 was remarkably co-localized with PDGFRβ in neovascular tufts and the CNV lesion (Fig. 1b, d), confirming that PDGFRβ was mainly expressed by pericytes.

**Preparation and characterization of ADC targeting mPDGFRβ in vitro**. Anti-mPDGFRβ × cotinine was expressed using a eukaryotic expression system and purified by affinity column chromatography. Before testing its internalization into mouse pericytes, we first confirmed by immunoblotting that mouse brain vascular pericytes (MBVPs) express a high level of PDGFRβ (Supplementary Figs. 1 and 2). Next, we tested the reactivity of the fusion protein to PDGFRβ in MBVP cells by flow cytometry (Fig. 2a). Thereafter, we performed confocal microscopy using allophycocyanin (APC)-conjugated anti-human C$_κ$ antibody to evaluate internalization of the fusion protein into MBVPs (Fig. 2b). In parallel, the cells were also stained with anti-Rab5 reactive to endosomes. When the two images were merged, anti-mPDGFRβ × cotinine was noted to co-localize with the endosome-specific antibody, confirming that the fusion protein

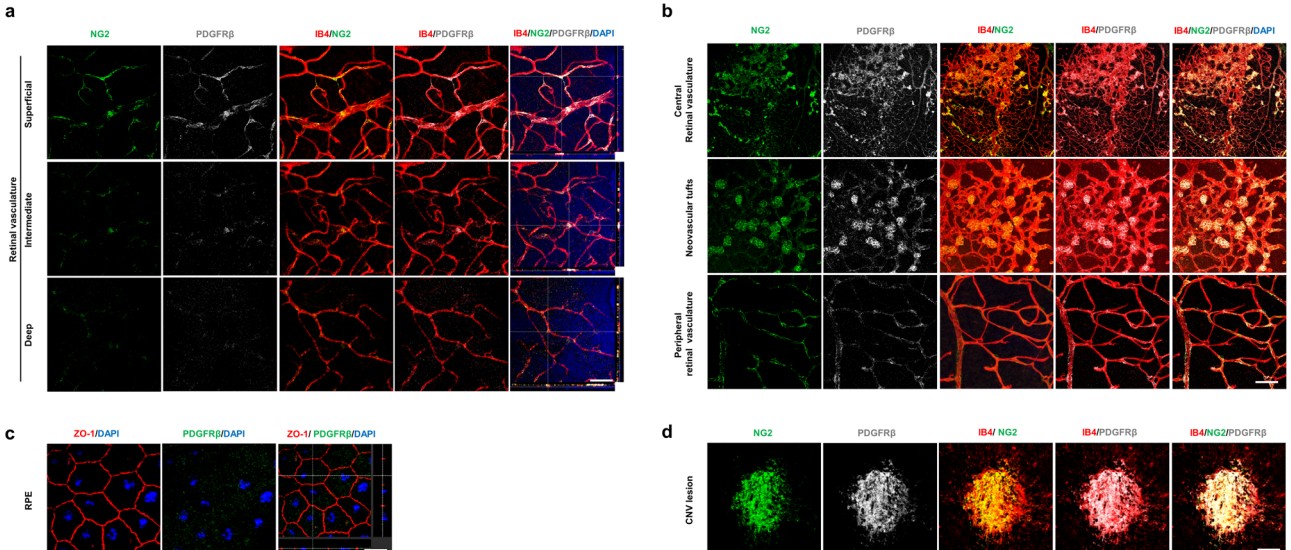

**Fig. 1 PDGFRβ is highly expressed on pericytes in pathologic neovessels of OIR and LI-CNV.** Representative immunofluorescence images of the retinal vasculature in a wild-type (control) mouse (**a**) and an oxygen-induced retinopathy (OIR) mouse (**b**). Immunofluorescence images of anti-isolectin B4 (IB4, red), anti-neural/glial antigen 2 (NG2, green), platelet-derived growth factor receptor β (PDGFRβ, gray), and 4′,6-diamidino-2-phenylindole (DAPI, blue) showing that PDGFRβ expression is mainly localized in the retinal vessels and surrounding tissue. In the OIR mouse, PDGFRβ is significantly overexpressed in the vascular tufts (**b**). Representative immunofluorescence images of retinal pigment epithelium (RPE) in a wild-type (control) mouse (**c**) and a mouse with laser-induced choroidal neovascularization (LI-CNV) (**d**). The choroidal neovascularization (CNV) lesion in the LI-CNV mouse shows strong PDGFRβ expression (**d**). Scale bar, 200 μm.

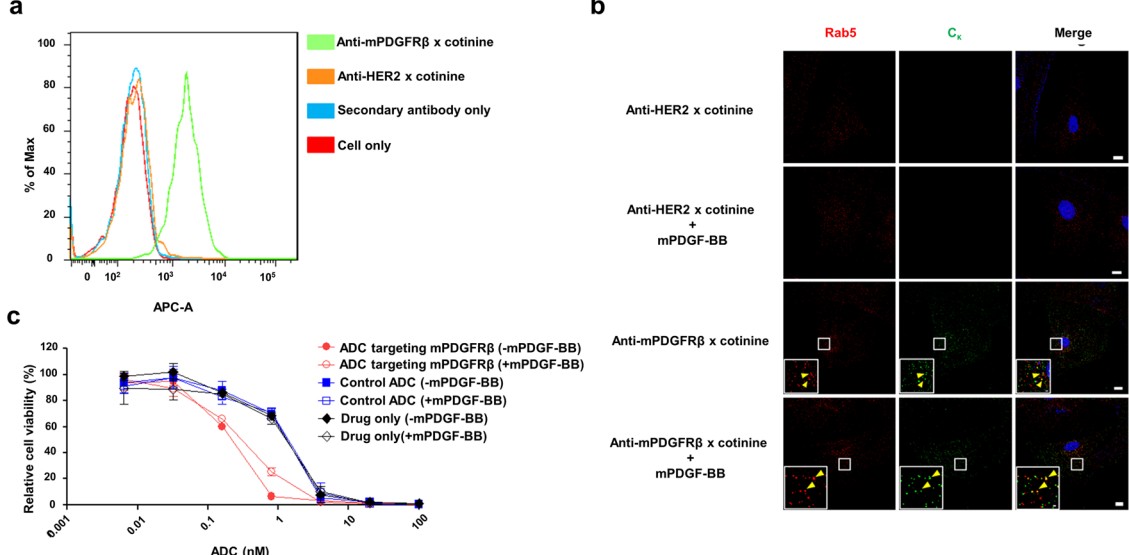

**Fig. 2 Preparation and characterization of ADC targeting mPDGFRβ. a** Flow cytometry of bispecific anti-mouse platelet-derived growth factor receptor β (mPDGFRβ) × cotinine single-chain variable fragment (scFv)- kappa constant (C$_\kappa$)-scFv fusion proteins (anti-mPDGFRβ × cotinine). Mouse brain vascular pericytes (MBVPs) were treated with anti-mPDGFRβ × cotinine and probed with APC-conjugated anti-human C$_\kappa$ antibody. As a control, anti-human epidermal growth factor receptor 2 (HER2) × cotinine scFv-C$_\kappa$-scFv fusion protein (anti-HER2 × cotinine) was used; anti-HER scFv was derived from trastuzumab. **b** Confocal microscopy of the internalization of anti-mPDGFRβ × cotinine. MBVP cells were incubated with scFv-C$_\kappa$-scFv fusion proteins with or without mPDGF-BB. Scale bars in the inset image and on the bottom right correspond to 1 μm and 10 μm, respectively. **c** Cytotoxicity of antibody-drug conjugate (ADC) targeting mPDGFRβ. MBVP cells were treated with ADC in the absence (−) or presence (+) of mPDGF-BB. The relative cell viability was calculated using the cellular ATP level. Anti-HER2 × cotinine complexed with cotinine-duocarmycin was used as a control ADC and drug only refers to cotinine-duocarmycin in fresh media. The results are shown as means ± standard deviation (SD) from triplicate experiments.

was internalized via the classical endocytosis pathway. We also mimicked the typical ocular NV environment with increased PDGF-BB levels by adding mouse PDGF-BB (mPDGF-BB) to the culture medium and observed no effects on internalization of the fusion protein. Cytotoxic effects of ADC targeting mPDGFRβ were tested in an in vitro setting. MBVP cells were cultured with the ADC for 72 h in the presence or absence of mPDGF-BB. Next, the cellular ATP content was measured to determine cell viability.

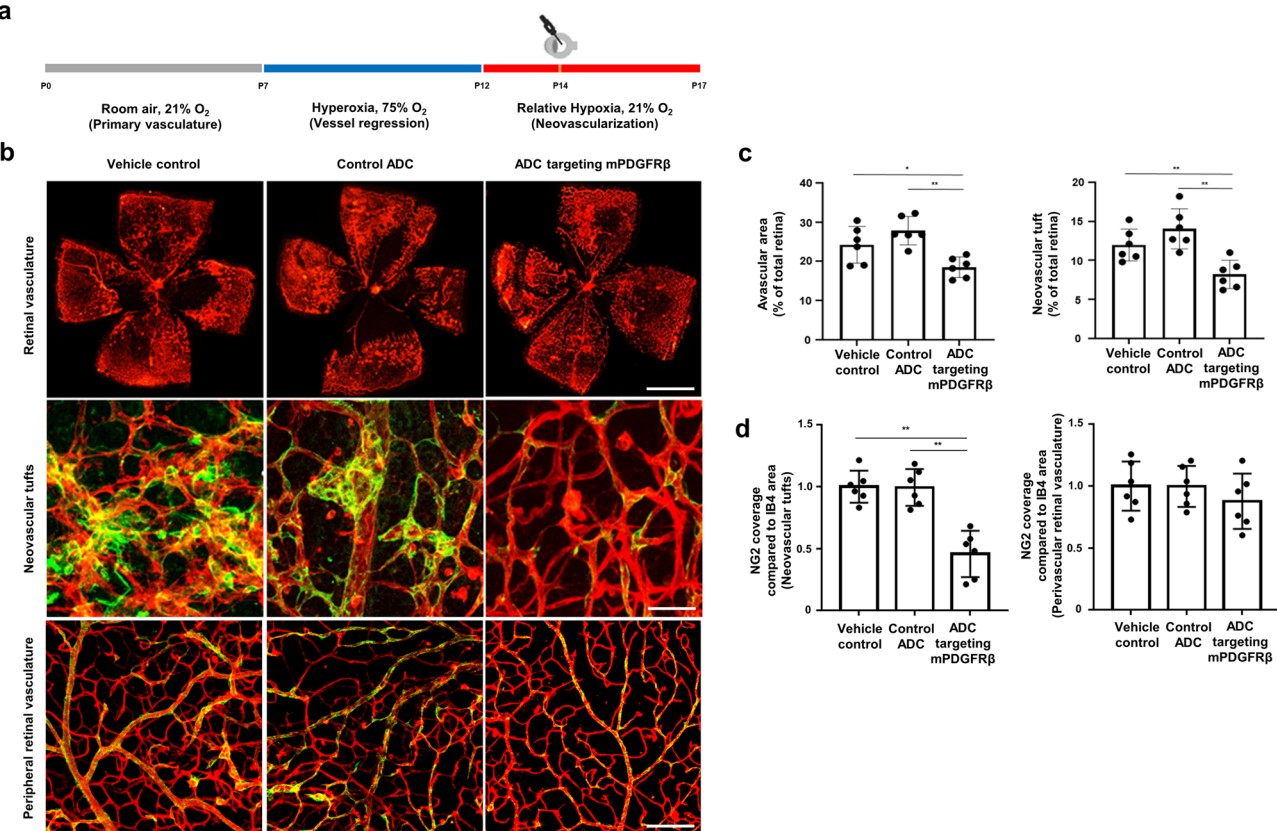

**Fig. 3 ADC targeting mPDGFRβ ameliorates retinal neovascularization in the OIR model. a** Schematic experimental timeline of antibody-drug conjugate (ADC) administration and tissue preparation in the oxygen-induced retinopathy (OIR) mouse model. OIR-induced pups were treated with vehicle control, control ADC, or ADC targeting mouse platelet-derived growth factor receptor β (mPDGFRβ) through intravitreal injection at P14. Three days after treatment, the mouse eyeballs were dissected to prepare whole-mounted retina samples. Phosphate-buffered saline (PBS) was used as the vehicle control. Anti-human epidermal growth factor receptor 2 (HER2) × cotinine complexed with cotinine-duocarmycin was used as a control ADC; anti-HER2 scFv was derived from trastuzumab. **b** Representative immunofluorescence images of whole-mounted retina samples from OIR pups stained with isolectin B4 (IB4, red) and neural/glial antigen 2 (NG2, green) to visualize the retinal vessels and pericytes. Scale bars on the top, middle, and bottom correspond to 500 μm, 100 μm, and 150 μm, respectively. **c** Quantification of the avascular area and neovascular tuft area. The avascular area and neovascular tuft area were quantified and presented as percentages of the total area of the retina. All data were analyzed using NIH ImageJ software, and values are presented as percentages of the mean ± SEM ($n = 6$ mice for each data set). *$P < 0.01$, **$P < 0.001$, obtained using one-way analysis of variance (ANOVA) and Tukey's post-hoc tests. **d** Quantification of NG2 coverage of IB4 + vessels at the mid-peripheral region (pathological neovascular tuft area, left) and peripheral region (peripheral vascularized area, right), respectively in the whole-mounted retinas. All data were analyzed using a built-in measuring tool, the LAS X system (Leica Microsystems Ltd., Wetzlar, Germany). Error bars represent standard error mean (SEM, $n = 6$ mice for each data set). **$P < 0.001$, obtained using one-way ANOVA and Tukey's post-hoc tests. Source data are provided as a Supplementary Data file.

Anti-mPDGFRβ ADC showed a half maximal inhibitory concentration value of 0.19 nM and statistically significant cytotoxicity, compared with the control ADC ($P < 0.001$; Fig. 2c and Supplementary Table 1). Addition of mPDGF-BB slightly increased the half maximal inhibitory concentration to 0.3 nM; this increase was not statistically significant ($P = 0.2898$; Fig. 2c and Supplementary Table 1).

**ADC targeting mPDGFRβ significantly inhibits NV in OIR and LI-CNV models.** In the OIR model, newborn C57BL/6J wild-type mice were exposed to hyperoxia between postnatal day 7 (P7) and postnatal day 12 (P12) to suppress physiologic vascular development. When returned to room air on P12, the relative retinal hypoxia led to retinal NV, which began between P12 and postnatal day 17 (P17) (Fig. 3a). ADC targeting mPDGFRβ (66.77 pg) was injected into the vitreous cavity in six mice on postnatal day 14 (P14). In parallel experiments, two groups of six mice were injected with either PBS or control ADC (66.77 pg) on the same day. On P17, the retina was retrieved from the mice for

whole-mounting and stained with fluorescein-labeled antibodies against IB4 and NG2, specific markers of vascular endothelial cells and pericytes, respectively. Areas of the avascular zone and vascular tuft were dramatically reduced in the retinas of mice treated with ADC targeting mPDGFRβ, compared with the retinas of mice injected with vehicle control or control ADC; the differences were statistically significance ($P < 0.01$; Fig. 3b, c). The expression level of NG2 at neovascular tuft was also dramatically decreased in mice treated with ADC targeting mPDGFRβ, indicating a significant reduction in the number of pericytes ($P < 0.01$; Fig. 3b, d), whereas no significant differences were found in NG2 expression at peripheral vascularized retina among the ADC group and two control groups ($P = 0.482$; Fig. 3b, d).

In the LI-CNV model, laser photocoagulation was performed on 6-week-old wild-type C57BL/6J male mice, rupturing Bruch's membrane (the innermost layer of the choroid) to induce CNV. Seven days later, ADC targeting mPDGFRβ (667.7 pg) was intravitreally injected into six mice. In parallel experiments, two groups of six mice were injected with PBS or control ADC (667.7 pg). At 10 days after laser photocoagulation, RPE-choroid-

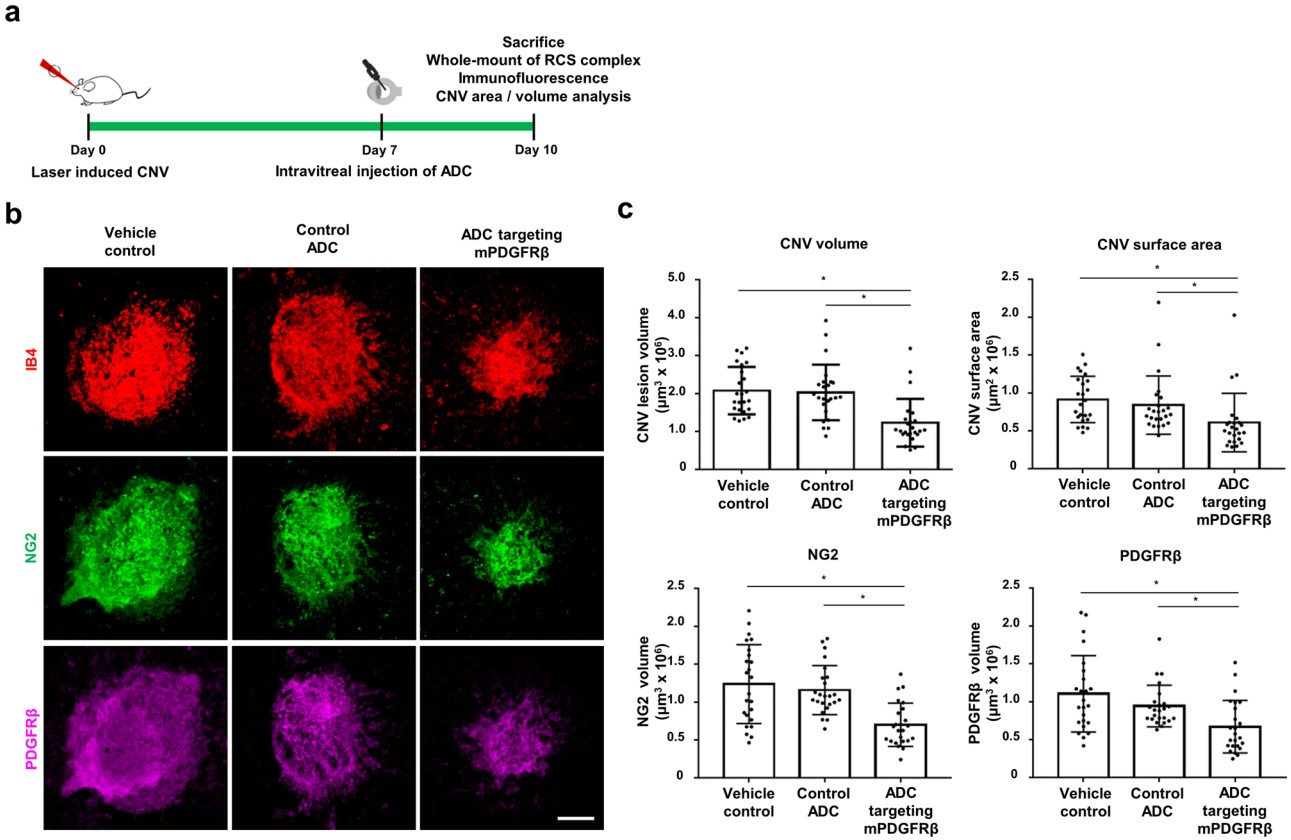

**Fig. 4 ADC targeting mPDGFRβ inhibits laser-induced CNV. a** Schematic experimental timeline of ADC administration and tissue preparation in the laser induced-choroidal neovascularization (LI-CNV) mouse model. Six-week-old wild-type C57BL/6J male mice received laser photocoagulation and were treated with vehicle control, control antibody-drug conjugate (ADC), or ADC targeting mouse platelet-derived growth factor receptor β (mPDGFRβ) through intravitreal injection at 7 days. Three days after treatment, the mouse eyeballs were dissected, and RPE-choroid-scleral (RCS) complexes were whole mounted. Phosphate-buffered saline (PBS) was used as the vehicle control. Anti-human epidermal growth factor receptor-2 (HER2) × cotinine complexed with cotinine-duocarmycin was used as a control ADC; anti-HER2 scFv was derived from trastuzumab. **b** Representative images of CNV at 10 days after laser photocoagulation and immunostaining with isolectin B4 (IB4), neural/glial antigen 2 (NG2), and PDGFRβ. Scale bar, 200 μm. **c** Quantitation of CNV volume, CNV area, NG2 volume, and PDGFRβ volume. All quantitative data were measured using the built-in tools, the LAS X systems (Leica Microsystems, Wetzlar, Germany). Each value represents the mean ± standard error mean (SEM, n = 6 mice for each group set). *P < 0.01, obtained using one-way analysis of variance (ANOVA) and Tukey's post-hoc tests. Source data are provided as a Supplementary Data file.

sclera (RCS) complexes were retrieved from the mice for mounting and then stained with fluorescein-labeled antibodies against IB4, NG2, or PDGFRβ (Fig. 4a). The volume and area of CNV and the volume of NG2- and PDGFRβ-positive tissue were markedly decreased in mice treated with ADC targeting mPDGFRβ, compared with mice administered PBS or control ADC (P < 0.01; Fig. 4b, c).

In the additional experiments comparing ADC targeting mPDGFRβ with conventional anti-angiogenic agent (1 μg/eye, bevacizumab, Avastin®; Genentech Inc., San Francisco, CA) of OIR and LI-CNV model, avascular area and neovascular tuft area were significantly reduced in the retina injected with bevacizumab and ADC targeting mPDGFRβ compared to vehicle control (P < 0.01; Supplementary Fig. 3a, b). Both bevacizumab and ADC targeting mPDGFRβ treatment significantly reduced NG2- and PDGFRβ positive area at neovascular tufts compared to vehicle control (P < 0.01; Supplementary Fig. 3c, d), whereas there were no significant differences in NG2 and PDGFRβ positive area at peripheral retinal vasculature among three groups (P = 0.269, P = 0.509; Supplementary Fig. 3e, f). These results suggest that pericytes in the neovascular tuft are more vulnerable than pericytes in the peripheral retinal vasculature against ADC targeting mPDGFRβ treatment. And the volume and area of CNV

and the volume of NG2- and PDGFRβ positive tissue were significantly decreased in mice treated with anti-VEGF antibody and ADC targeting mPDGFRβ, compared to mice treated with vehicle control (P < 0.01; Supplementary Fig. 4).

**ADC targeting mPDGFRβ does not induce anatomic or functional toxicity in the retina.** To evaluate whether ADC targeting mPDGFRβ induces anatomic toxicity in the retina, ADC (667.7 pg) was administered intravitreally into 6-week-old male wild-type C57BL/6J mice. PBS or control ADC was injected on the same day to mice in the control groups. Seven days after injection, the eyeballs were enucleated, fixed in 4% paraformaldehyde, and embedded in paraffin. Next, tissue sections were prepared and stained with H&E. Retinal layer thickness was determined by calculating the ratio of A (retinal thickness from the internal limiting layer to the inner nuclear layer) to B (retinal thickness from the internal limiting layer to the outer nuclear layer) in the section images captured by the light microscope. To check the degree of apoptosis, the TUNEL assay was performed, and the number of positive cells was determined at ×200 magnification under a fluorescence microscope in five randomly selected fields in each slide. No significant differences were found

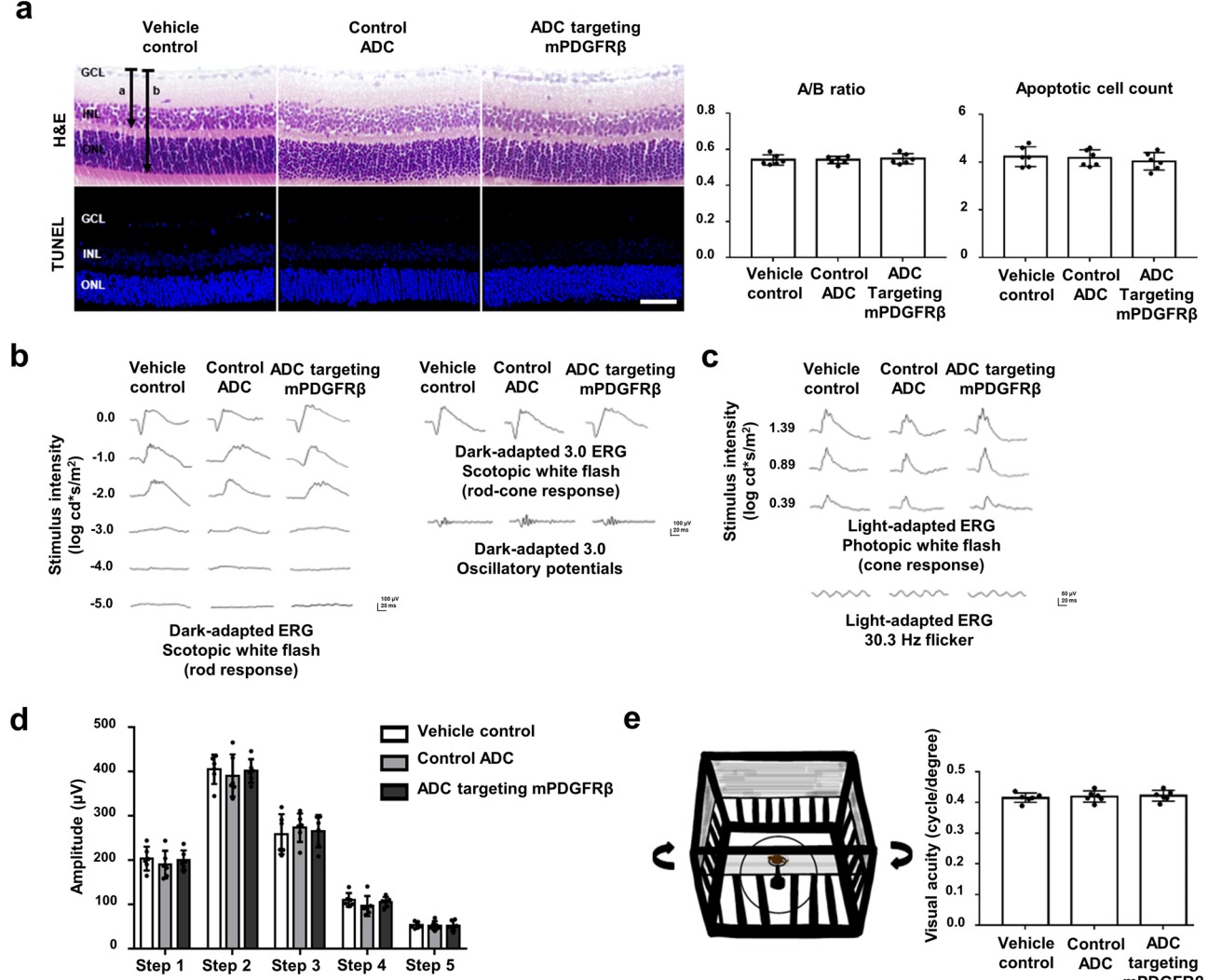

**Fig. 5 Retinal safety evaluation in mice receiving ADC targeting mPDGFRβ. a** Representative hematoxylin and eosin (H&E) and terminal deoxynucleotidyl transferase dUTP nick end labeling (TUNEL) staining images of retinal tissues at 7 days after injection of vehicle control, control antibody-drug conjugate (ADC), or ADC targeting mouse platelet-derived growth factor receptor β (mPDGFRβ). Scale bar, 500 μm. Phosphate-buffered saline (PBS) was used as the vehicle control. Anti-human epidermal growth factor receptor 2 (HER2) × cotinine complexed with cotinine-duocarmycin was used as a control ADC; anti-HER2 single-chain variable fragment (scFv) was derived from trastuzumab. Each value represents the mean ± standard error mean (SEM, $n = 6$ mice per data set). **b** The overall stimulus parameters used in the study for the representative group of averaged dark-adapted electroretinography (ERG) waveforms were as follows: step 1: scotopic white flash; step 2: scotopic white flash 3.0 cd·s/m²; and step 3: oscillatory potential, 3.0 cd·s/m². **c** The overall stimulus parameters used in the study for the representative group-averaged light-adapted ERG waveforms were as follows: step 4: photopic white flash; and step 5: 30.3 Hz flicker. **d** Amplitudes of the b-wave by steps at 7 days after the injection of PBS, control ADC, or ADC targeting mPDGFRβ. Anti-HER2 × cotinine complexed with cotinine-duocarmycin was used as a control ADC; anti-HER2 scFv was derived from trastuzumab. Each value represents the mean ± standard error mean (SEM, $n = 6$ mice per data set). **e** Schematic diagram depicting the OptoMotry test and spatial frequency thresholds, in cycles per degree. Each value represents the mean ± SEM ($n = 6$ mice per data set). No significant differences were found in the retinal layer thickness ratio, terminal deoxynucleotidyl transferase dUTP nick end labeling (TUNEL) assay, ERG amplitudes, or spatial frequency of the OptoMotry test among the three groups, as confirmed by one-way analysis of variance (ANOVA) and Tukey's post-hoc tests. Source data are provided as a Supplementary Data file.

in retinal thickness or number of apoptotic cells between the ADC group and two control groups (Fig. 5a).

To monitor functional toxicity 7 days after injection of ADC targeting mPDGFRβ, we performed ERG and the OptoMotry test. To assess general retinal function, full-field ERG was conducted under light- and dark-adapted conditions. After dark adaptation for over 16 h, dark-adapted ERG was recorded while projecting series of white flashes of increasing stimulus strength within a Ganzfeld dome (Fig. 5b). Next, the mice were exposed to a steady field of light for 15 min to maximize the cone response and minimize the rod input, and then we recorded the light-adapted ERG while a series of white flashes of increasing stimulus strength

was projected (Fig. 5c). We analyzed the b-wave amplitude (step 1, 2), oscillatory potential amplitude (step 3) of the scotopic response, and b-wave amplitude (step 4), and flicker amplitude (step 5) of the photopic response, which are major components of the ERG response. The b-wave was measured from the a-wave trough (a-wave amplitude: from baseline to the maximum negative trough) to the maximum positive peak. No significant differences were detected in these amplitudes between the ADC targeting mPDGFRβ group and the two control groups ($P = 0.983$; Fig. 5d). To measure the spatial frequency threshold, homogeneous sine-wave gratings (100% contrast) of the same mean luminance were projected while rotating in one direction

(counterclockwise or clockwise). The mice were assessed for tracking behavior, and then the spatial frequency of the sine-wave gratings was adjusted following the optokinetic response of the mice. This procedure was systematically repeated until the animal no longer responded, indicating the threshold point. No significant differences were found in the spatial frequency threshold between the ADC targeting mPDGFRβ group and two control groups ($P = 0.284$; Fig. 5e). Thus, ADC targeting mPDGFRβ does not alter anatomic or functional retinal integrity.

## Discussion

Anti-VEGF agents are standard therapeutics for ocular NV, reducing the deterioration of visual acuity and improving subjective visual symptoms. But repeated intravitreal injections of anti-VEGF agents lead to intraocular complications, such as fibrosis, hemorrhage, and atrophy of the retina and choroid[12–18]. Additionally, some patients are resistant or refractory to anti-VEGF reagents. Therefore, there are still needs for the development of novel therapeutics for ocular NV, either as stand-alone agents or in combination with anti-VEGF reagents. In this study, we demonstrated that an ADC targeting PDGFRβ selectively ablated PDGFRβ-expressing cells and inhibited pathologic ocular NV in mouse models of OIR and LI-CNV.

During vascular development, endothelial cells release PDGF-BB, most of which binds to PDGFRβ in perivascular cells[54]. PDGF-BB facilitates vessel maturation by promoting pericyte recruitment and supporting pericyte-endothelial interactions. The recruitment of pericytes is dependent on PDGF-BB/PDGFRβ signaling. In the absence of PDGF-BB or PDGFRβ, near-complete loss of pericytes is observed in organs such as those in the central nervous system[55]. In PDGF-BB-deficient mouse embryos, endothelial cells of sprouting capillaries could not attract PDGFRβ-positive pericyte progenitor cells, leading to numerous capillary microaneurysms without a stable capillary wall[22]. Pericytes directly contact vascular endothelial cells, sharing a common basement membrane. This biological trait provides extra stability to vascular endothelial cells, preventing vascular leakage. Pericytes also secrete VEGF and other cell survival factors to protect vascular endothelial cells[23,29,30]. These protective functions of pericytes on endothelial cells render neovascular endothelial cells resistant to anti-VEGF therapy[31,34].

Upregulated intraocular levels of PDGF-BB have been reported in patients with proliferative diabetic retinopathy and CNV[56,57] as well as in OIR and LI-CNV models[58–61]. Furthermore, an inverse correlation has been observed between concentrations of PDGF/PDGFRβ and VEGF during anti-VEGF treatments in patients with AMD[40–42]. Other reports described excessive coverage of PDGFRβ-positive cells in the NV region in OIR and LI-CNV[37,58–61]. Furthermore, in a patient with bilateral CNV, the neovascular membrane of the eye unresponsive to bevacizumab exhibited increased numbers of pericytes, whereas the eye that responded to treatment had a limited number of these cells[43].

In preclinical studies, pegpleranib, an aptamer that binds to PDGF-AB and PDGF-BB and inhibits their interaction with PDGFRα and PDGFRβ, showed efficacy in inhibiting CNV[37], and an anti-PDGFRβ antibody, rinucumab, also inhibited mouse CNV. However, neither pegpleranib nor rinucumab have exhibited additive therapeutic effects when used in combination with anti-VEGF agents in clinical trials[39]. A few small-molecule tyrosine kinase inhibitors of both VEGF and PDGF receptors, including DE-120 and vorolanib, have been tested clinically. However, none of these agents are currently available in clinical settings. Hence, we considered the use of ADC that directly targets PDGFRβ-expressing cells and selectively induces cell death as an alternative therapeutic strategy.

Interestingly, affibody-IR700 dye conjugate targeting PDGFRβ is predominantly distributed on pericytes inside tumors and induces tumor destruction via photodynamic therapy, with accompanying angiogenesis inhibition and tissue hypoxia[62,63], suggesting the possibility of removing pericytes using ADC. For ADC to be effective, a sufficient number of target molecules should be present on the target cell surface and internalized efficiently. To avoid toxicity, the antigen should be preferentially expressed on the surface of target cells, compared with normal cells[64,65]. PDGFRβ is mainly expressed on pericytes[26,27]. In our experiments, PDGFRβ was significantly overexpressed in the NV region, compared with the control retina and RPE, in both the OIR and LI-CNV models (Fig. 1).

Based on these observations, we hypothesized that ADC targeting PDGFRβ might induce cytotoxicity specifically toward pericytes of neovessels and inhibit ocular NV. In prior studies, we developed a unique form of ADC consisting of an anti-EGFR × cotinine bispecific antibody and cotinine-duocarmycin conjugate[50]. This ADC showed significant in vivo antitumor activity against EGFR-positive lung adenocarcinoma cells. We subsequently adopted this ADC platform to produce an ADC consisting of anti-mPDGFRβ × cotinine and cotinine-duocarmycin conjugate and confirmed that it could induce cytotoxicity specifically toward PDGFRβ-expressing cells. Notably, we selected an anti-PDGFRβ antibody that does not compete with PDGF-BB in its binding to PDGFRβ[51]. We have previously constructed cotinine-conjugated valine-citrulline-PAB-maleimidomethyl cyclohexane-1-carboxylate (mcc)-mertansine (DM1), valine-citrulline-PAB-Monomethyl auristatin E (MMAE), and valine-citrulline-PAB-duocarmycin. We have tested and discovered cotinine-duocarmycin conjugate was the most potent and that drug-to-antibody ratios (DAR)4 cotinine-cytotoxic drugs were more potent than DAR1 cotinine-cytotoxic drugs. However, the ADC was not purified after the conjugation and average DAR was not determined. Hence, the average DAR of the ADC may be lower than the DAR of cotinine-cytotoxic drugs. Furthermore, duocarmycin being hydrophilic, minimizes by-stander effect and hence less non-specific off-target effects[66]. For these reasons, we have only used cotinine-duocarmycin for detailed studies. In the current study, we first confirmed that ADC targeting mPDGFRβ efficiently killed MBVP cells and verified that its internalization and cytotoxicity were not affected by the presence of PDGF-BB in an in vitro setting (Fig. 2). We then established OIR and LI-CNV mouse models and showed that ADC targeting mPDGFRβ remarkably inhibited progression of the avascular area, neovascular tufts (Fig. 3), and CNV lesions (Fig. 4).

In the OIR model, we also found that ADC targeting mPDGFRβ could promote retinal revascularization, as well as ameliorate neovascular tufts. Under the ischemic retinal conditions of OIR, neovascular tufts covered by pericytes grow toward the vitreous body[61,67,68]. In our study, both bevacizumab and ADC targeting mPDGFRβ treatment significantly reduced NG2- and PDGFRβ positive area compared to vehicle control at the boundary between the vascular and avascular areas of the retina, whereas there were no differences of NG2 and PDGFRβ positive area at the peripheral vascularized retina. We considered that pericytes in the neovascular tufts are more vulnerable than pericytes in the peripheral retinal vasculature from these data (Supplementary Fig. 3). This result is likely due to overexpression of PDGFRβ of the pericytes in the neovascular tufts than pericytes in the peripheral retinal vasculature, increasing efficiency of ADC targeting mPDGFRβ. In other study, pericyte-specific deletion of CCN1, an extracellular matrix protein, during angiogenesis under ischemic conditions results in reduced neovascular tufts and increased revascularization[67]. Another group reported that

NCK1/2 signaling downstream of PDGFRβ in pericytes was related to OIR progression, and its inhibition suppressed pericyte recruitment to the tip cells, preventing neovascular tuft formation and promoting revascularization[61]. These findings suggest an important role of pericytes in the progression of normal angiogenesis, as well as in the regression of neovascularization in OIR.

Immunohistochemical experiments showed a marked difference in expression levels of PDGFRβ between normal retinal vessels and neovessels (Fig. 1), which would limit the toxicity of ADC targeting mPDGFRβ toward normal retinal vessels. This lack of toxicity by ADC targeting mPDGFRβ was confirmed in both anatomic and functional aspects. The retinal layer thickness ratio was not changed, and TUNEL assays were negative (Fig. 5). The results of ERG and the OptoMotry test were also normal. To further evaluate retinal vessel toxicity, intravitreal injection of different doses of ADC targeting mPDGFRβ (dose range: 667.7 pg ~ 333.85 ng per eye) was administered to 6-week-old wild-type C57BL/6J male mice. Although treatment of ADC targeting mPDGFRβ up to 100 times higher dose than therapeutic dose in the LI-CNV model did not affect retinal integrity, treatment of ADC targeting mPDGFRβ of 500 times higher dose induced a significant reduction in the number of pericytes and inner blood-retinal barrier leakage (Supplementary Fig. 5). There has been reported that a sudden reduction of pericytes from stabilized vessels is not sufficient to disrupt the integrity of retinal vessels[69]; limited toxicity toward pericytes in normal vessels could be endured without causing pathologic changes in the retina. According to our toxicity test by different doses of ADC, we require meticulous evaluation to minimize the potential retinal vessel toxicity when determining the optimal dose of ADC targeting mPDGFRβ for clinical trials. Since the ADC was not purified after the conjugation, unpurified payload could have contributed to non-specific killing. However, considering systemic toxicity, the risk would be very minimal because the amount of ADC administered is quite limited, and the ADC is injected locally via the intravitreal route. In our experiments, mice injected with ADC targeting mPDGFRβ showed no behavioral change or weight loss (Supplementary Table 2).

Regarding the systemic clearance of ADC targeting mPDGFRβ, when injected intravenously, the in vivo half-life of the anti-cotinine antibody and cotinine-conjugate complex was 15 h[70], and that of the scFv-$C_\kappa$-scFv molecule was ~6 h (manuscript under preparation). It was reported that ranibizumab (48 kDa) with a half life of 2.88 days showed faster penetration and elimination from retina than bevacizumab (149 kDa) with a half life of 4.32 days due to smaller size[71]. Because the molecular weight of the cotinine-duocarmycin conjugate is 6.27 kDa, it would be rapidly excreted through the kidney after being released from anti-mPDGFRβ × cotinine. We selected cotinine as a hapten to conjugate duocarmycin because of its non-toxic nature, with a median lethal dose of $4 \pm 0.1$ g/kg in mice[48], as well as its absence in physiologic situations and relative pharmacologic inertness[72]. The metabolic pathways of duocarmycin[73] and cotinine[52] were previously analyzed in detail.

In conclusion, removal of PDGFRβ-overexpressing pericytes in neovessels with ADC effectively suppressed pathologic ocular NV in OIR and LI-CNV mouse models. At the therapeutic dose, ADC targeting PDGFRβ did not induce morphologic or functional abnormalities in the retina or cause systemic toxicity. We believe that ADC targeting PDGFRβ can be a therapeutic option for ocular NV after repeated clinical failures of PDGF/PDGFR signaling inhibitors.

## Data availability

The authors declare that the main data supporting the results in this study are included within the paper and its Supplementary Information files. The microscopy data for

in vivo are deposited on zenodo (https://doi.org/10.5281/zenodo.5598741)[74]. Source data for the main figures in the manuscript can be found in "Supplementary Data 1".

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

## Acknowledgements

This work was supported by a grant from the National Research Foundation funded by the Korean government (NRF-2018M3D1A1058826, NRF-2015M3A7B6027946, KRISS-2020-GP200-0004, NRF-2017M3A9C8032204, and NRF-2019R1A6C1010020).

## Author contributions

J.H.K., J.Chung, and D.H.J. designed the study and supervised the experiments. S.R.K. and D.K. contributed to confocal imaging in vitro. S.K., J.Chae, D.K.Y., and S.H. performed the rest of the in vitro experiments and analyzed the data. C.S.C. contributed to the OIR experiment. S.J.L. performed the rest of in vivo experiments and analyzed the data. S.J.L., S.K., J.Chung, and J.H.K. wrote the paper and received input from all coauthors.

## Competing interests

The authors declare no competing interests.
