## [Peer Review File · Communications Medicine]

Reviewers' comments:

Reviewer #1 (Remarks to the Author):

In the manuscript by Seok Jae Lee et al., the authors demonstrated that specific ablation of PDGFR β -overexpressing pericytes using an antibody-drug conjugate (ADC) potently inhibited pathologic ocular neovascularization in mouse models of oxygen-induced retinopathy and laser-induced choroidal neovascularization. Given that most blinding eye diseases have a neovascular component, another potent anti-angiogenic would be a useful adjunct to current therapies that largely focus on targeting VEGF.

Several points need to be clarified before the manuscript would be suitable for publication.

1. Figure1.

Pink (PDGFR β) and red (IB4) are difficult to distinguish from one another. As the expression of PDGFR β and its signal intensity in retinal vessels and neovascularization are essential findings for this manuscript, the authors should show single staining images for PDGFR β and NG-2 as well as merged images with IB4 using colors that can be distinguished from one another.

2. Figure3.

The authors showed only an NG2 staining image for each treatment in neovascular tufts after ADC or control injections. As the number of NG2 positive cells are essential to demonstrate how much ADC targeting mPDGFR β is effective and safe, the authors should quantify NG2 positive cells in both neovascular areas and vascularized area for each treatment.

3. As the authors describe, pericytes have critical physiological functions for stabilizing endothelial cells and preventing leakages consisting of the blood-retinal barrier. The authors should evaluate the number of pericytes and vascular leakage after ADC injection into wild-type mouse eyes.

4. Both the OIR and laser-induced CNV models are notoriously variable and it seems that an n of 6 for each variable done as a single experiment would not be adequate to have confidence in the efficacy of this approach. It would also be helpful to show the effects of other anti-angiogenics (e.g. a VEGF antagonists, the gold standard of therapy today) in the investigators' hands and models given that the reduction in NV and vaso-obliteration after targeting mPDGFR β is marginal and of a value that would not nearly eliminate this pathology.

Reviewer #2 (Remarks to the Author):

Lee et al. describe the effect of a bivalent anti-PDGFR β , anti-cotinine antibody conjugated to cotinine-duocarmycin and delivered by intravitreal injection in mouse P14 pups after they were exposed to high oxygen to induce retinopathy, or to laser-induced choroidal neovascularization in the Bruch's membrane. In both conditions, treatment with the antibody decreased the area of abnormal vasculature. This is a study with novel data describing a reagent with potential clinical relevance for treatment of common ocular diseases.

1. In their previous article in *Methods* (2019), the authors describe how the bispecific antibody was designed. This needs to be briefly reiterated in the current presentation along with an explanation for why it is advantageous to use anti-cytinine for conjugation of the cytotoxic drug. Also please give the rationale for choosing duocarmycin as the cytotoxic drug.
2. In Fig. 3A, please complement the panel with image and quantification of pericyte area and avascular/tuft area in the oxygen-challenged retina at P14 (at the point of treatment), to understand the effect of the antibody treatment. Does the antibody treatment arrest pericyte and tuft morphology at their P14 size?
3. In Fig. 3A, are pericytes undergoing apoptosis after treatment with the bivalent anti-PDGFRb antibody? Please show PDGFRb-positive area in the different treatment conditions, normalized to total vascular area. Examine pericytes both in tufts and in the remaining vascularized area to show whether PDGFRb-positive cells in the tufts are more vulnerable than pericytes associated with the remaining, unaffected vasculature.
4. The authors must have carefully titrated the optimal dose to use for treatment as a higher dose would strip pericytes completely from the retina vasculature, which would cause vascular abnormalities in itself. Please explain and preferably, show the effect of different doses of the bivalent anti-mPDGFR drug-conjugated antibody.
5. Is inflammation in the retina increased with the antibody treatments?

Minor

6. Instead of writing the full name of the bivalent anti-mPDGFRb antibody each time it is mentioned, perhaps the authors could give a short abbreviation.
7. In Fig 1, it is difficult to see the difference in the colors chosen for PDGFRb (pink) and IB4 (red). Please select other colors.
8. In Fig 1, panel c, in the list of treatments it says mPDGFR-BB which should be mPDGF-BB.

Reviewer #3 (Remarks to the Author):

In this manuscript, the authors investigated whether eradication of pericytes overexpressing mPDGFR β using an ADC could be a promising alternative to conventional anti-VEGF antibody-based therapy for ocular neovascular diseases. The authors prepared and characterized an anti-mPDGFR β ADC equipped with duocarmycin following the method previously reported by them. They subsequently tested the conjugate for the ability to deplete mPDGFR β -expressing MBVP cells in vitro and to suppress retinal neovascularization in mouse models. As the authors concluded, the ADC appears to exhibit mPDGFR β -specific cytotoxicity leading to amelioration of pathologic neovascularization while showing marginal toxicity. ADCs have been developed and used mostly for cancer therapy and ones for non-oncology applications remain to be fully explored. Thus, I believe the manuscript fits the journal's scope and is of great interest to the readership. However, several critical issues regarding ADC preparation and study design were identified. These issues likely

dampen or obscure the technical soundness of the work, the clinical potential of the ADC, and the significance of authors' findings. Thus, I suggest the authors to adequately address the concerns listed below.

Major concerns:

1. A conventional anti-VEGF antibody should be tested as a monotherapy control or combination therapy with the ADC in the in vivo treatment studies. Although potential advantages of anti-mPDGFR β agents over anti-VEGF antibodies are described, the manuscript does not fully validate the clinical relevance and potential therapeutic benefits of the ADC due to lack of a comparison with this control group.
2. Was the ADC pure enough to be used for following in vitro/vivo testing? The authors prepared and characterized the ADC according to the authors' previous report (Ref 46) and apparently did not purify it after complexation with cotinine-duocarmycin. Supplementary Figure 3 of Ref 46 indicates that PRb-CN01 + cot-duo ADC (presumably the same ADC as the one used in this study) contained substantial amount of free cotinine-duocarmycin under physiological conditions. This unpurified payload could contribute to non-specific cell killing as observed for the control ADC in Fig 2c. More importantly, such a mixture may not be appropriate for clinical use. At least the authors should clarify this point and assure in the manuscript that the existence of the free payload does not compromise the study design and conclusion. Otherwise most studies should be performed with a purified ADC.
3. Fig 2c: although anti-mPDGFR β ADC with or without mPDGF-BB showed enhanced cell killing effect, both control ADC and vehicle also showed concentration-dependent cell killing in 1-10 nM range. Why did these groups, in particular vehicle control, show such unusually high toxicity?
4. The authors state that the in vivo half-lives of the conjugate and its component are 6-15 hours, which are much shorter than those of conventional ADCs (7-14 days). This indicates that frequent and multiple administrations may be required in clinical setting. The authors should discuss whether this shortcoming will negatively impact its practicality in future clinical use.

Minor concerns:

1. Line 104, Page 5: Several ADCs for non-oncology applications have been reported. Such reports should be cited (e.g., Nature 2015, 527:323-8; Bioconjugate Chem. 2018, 29:2357-2369)
2. Confocal microscopy: please report how many random images were used for quantification in each study.
3. Line 158, Page 7: please provide cell density of MBVP cells seeded on each well
4. Line 162, Page 7: please clearly describe what was in "0.006-500 nM" range. Final concentration of each ADC?
5. Body weight range of the mice at the beginning of each treatment study should be described.
6. Line 199, Page 9: were only male mice used as in the other animal studies?
7. Please add to Fig 2 or SI a schematic diagram showing the structures of the ADC and conjugation process. Also please provide the empirically determined drug-to-antibody ratio (DAR) and the method used.
8. Legend of Fig 2: where is "Scale bars on the bottom left"?
9. Line 480, Page 18: Please provide body weight data during this study. I do not see any reason for leaving out this simple information.

<Point-To-Point responses to reviewers>

Reviewer #1

In the manuscript by Seok Jae Lee et al., the authors demonstrated that specific ablation of PDGFR β -overexpressing pericytes using an antibody-drug conjugate (ADC) potently inhibited pathologic ocular neovascularization in mouse models of oxygen-induced retinopathy and laser-induced choroidal neovascularization. Given that most blinding eye diseases have a neovascular component, another potent anti-angiogenic would be a useful adjunct to current therapies that largely focus on targeting VEGF.

Several points need to be clarified before the manuscript would be suitable for publication.

1. Figure1.

Pink (PDGFR β) and red (IB4) are difficult to distinguish from one another. As the expression of PDGFR β and its signal intensity in retinal vessels and neovascularization are essential findings for this manuscript, the authors should show single staining images for PDGFR β and NG-2 as well as merged images with IB4 using colors that can be distinguished from one another.

→ We agree with the reviewer's suggestion. We added single staining images for PDGFR β and NG-2 and changed the color of PDGFR β (Pink to Gray) to make it easier to distinguish PDGFR β and IB4. (Fig. 1)

2. Figure3.

The authors showed only an NG2 staining image for each treatment in neovascular tufts after ADC or control injections. As the number of NG2 positive cells are essential to demonstrate how much ADC targeting mPDGFR β is effective and safe, the authors should quantify NG2 positive cells in both neovascular areas and vascularized area for each treatment.

→ We appreciate the reviewer's helpful comment. We added representative peripheral retinal vasculature images and quantified NG2 coverage compared to IB4 + vessels in the neovascular tuft and peripheral vascularized area for each treatment. And we added a description of quantification methods. (Page 10, line 217-222)

3. As the authors describe, pericytes have critical physiological functions for stabilizing endothelial cells and preventing leakages consisting of the blood-retinal barrier. The authors should evaluate the number of pericytes and vascular leakage after ADC injection into wild-type mouse eyes.

→ We fully agree with the reviewer's opinion that evaluating inner blood-retinal barrier status after ADC injection is needed. We intravitreally injected phosphate-buffered saline (as vehicle control) or different doses of ADC targeting mPDGFR β (667.7 pg [therapeutic dose in the LI-CNV model] ~ 333.85 ng [500 times higher dose than therapeutic does in the LI-CNV model] per eye) into 6-week-old male wild-type C57BL/6J mice. Seven days after injection, we conducted the blood-retinal barrier leakage test using FITC-dextran and quantified NG2 and PDGFR β coverage to IB4+ vessels following a single injection.

→ The results showed that the inner blood-retinal barrier in the ADC targeting mPDGFR β treatment group (dose range: 667.7pg ~ 66.77 ng [100 times higher dose than the therapeutic dose in the LI-CNV model] per eye) was intact, and there was no significant difference compared to the vehicle control group, whereas significant fluorescence leakage regions were detected in the ADC targeting mPDGFR β treatment group injecting 500 times higher dose than the therapeutic dose in LI-CNV model ($P < 0.01$; Supplementary Fig. 4a and 4b).

→ We also quantified NG2 and PDGFR β coverage to IB4+ vessels to assess pericytes loss. The NG2 and PDGFR β coverage to the retinal vessel was significantly decreased in the ADC targeting mPDGFR β treatment group (333.85 ng/eye) compared with the vehicle control group ($P < 0.01$; Supplementary Fig. 4c and 4d).

4. Both the OIR and laser-induced CNV models are notoriously variable and it seems that an n of 6 for each variable done as a single experiment would not be adequate to have confidence in the efficacy of this approach. It would also be helpful to show the effects of other anti-angiogenics (e.g. a VEGF antagonists, the gold standard of therapy today) in the investigators' hands and models given that the reduction in NV and vaso-obliteration after targeting mPDGFR β is marginal and of a value that would not nearly eliminate this pathology.

→ We appreciate the reviewer's comment. We newly conducted additional efficacy studies treating intravitreal injection of PBS (vehicle control), ADC targeting mPDGFR β , and anti-angiogenic agent (1 μ g/eye, bevacizumab, Avastin[®]; Genentech Inc., San Francisco, CA), which is well known as potent anti-angiogenic effects¹⁻⁴ without retinal toxicity^{5,6} in both OIR and laser-induced CNV mouse models and widely used as an off-label treatment for retinopathy of prematurity^{7,8} and age-related macular degeneration^{9,10}.

(1) *Clin Exp Ophthalmol.* 2019 Jan;47(1):79-87 (Comparison of anti-angiogenic effect among various anti-angiogenic treatments in OIR mouse model)

(2) *Int J Ophthalmol.* 2014 Aug 18;7(4):608-613 (OIR mouse model)

(3) *Curr Eye Res.* 2015 May;40(6):611-621 (OIR, LI-CNV mouse model)

(4) *Exp Eye Res.* 2021 Feb;203:108392. (LI-CNV mouse model)

(5) *Retina.* 2008;28(1):46-55 (Bevacizumab, toxicity test, mouse)

(6) *Neurotoxicology.* 2008;49(6):1131-1135 (Bevacizumab, toxicity test, mouse)

- (7) *Br J Ophthalmol.* 2008;92(11):1450-1455 (Retinopathy of prematurity)
(8) *Eur J Ophthalmol.* 2019 May;29(3):338-347 (Retinopathy of prematurity)
(9) *N Engl J Med.* 2011 May 19;364(20):1897-1908. (Age-related macular degeneration)
(10) *Ophthalmology.* 2012 Oct;119(10):2087-2093. (Age-related macular degeneration)

→ In the OIR model, avascular area and neovascular tuft were significantly reduced in the retina injected with bevacizumab and ADC targeting mPDGFR β compared to vehicle control ($P < 0.01$; Supplementary Fig. 2a and 2b). We also analyzed the expression level of NG2, PDGFR β at neovascular tuft (pathological angiogenesis area) and peripheral retinal vasculature (peripheral vascularized area). We found that both anti-angiogenic agent (bevacizumab) and ADC targeting mPDGFR β treatment significantly reduced NG2- and PDGFR β positive area at neovascular tufts compared to vehicle control ($P < 0.01$; Supplementary Fig. 2c and 2d), while there were no differences in NG2 and PDGFR β positive area at peripheral retinal vasculature among three groups ($P = 0.269$, $P = 0.509$; Supplementary Fig. 2e and 2f). These results suggest that the treating dosage of ADC targeting mPDGFR β in our OIR study (66.79 pg/eye) does not critically affect the peripheral vascularized area.

→ In the laser-induced CNV model, the volume and area of CNV and the volume of NG2- and PDGFR β positive tissue were significantly decreased in mice treated with the anti-angiogenic agent (1 $\mu\text{g}/\text{eye}$, bevacizumab) and ADC targeting mPDGFR β compared with mice administered PBS ($P < 0.01$; Supplementary Fig. 3).

Reviewer #2

Lee et al. describe the effect of a bivalent anti-PDGFRb, anti-cotinine antibody conjugated to cotinine-duocarmycin and delivered by intravitreal injection in mouse P14 pups after they were exposed to high oxygen to induce retinopathy, or to laser-induced choroidal neovascularization in the Bruch's membrane. In both conditions, treatment with the antibody decreased the area of abnormal vasculature. This is a study with novel data describing a reagent with potential clinical relevance for treatment of common ocular diseases.

1. In their previous article in Methods (2019), the authors describe how the bispecific antibody was designed. This needs to be briefly reiterated in the current presentation along with an explanation for why it is advantageous to use anti-cotinine for conjugation of the cytotoxic drug. Also please give the rationale for choosing duocarmycin as the cytotoxic drug.

→ Thank you for your thoughtful suggestion. We added descriptions of the antibody design and the advantages of using cotinine in the introduction (Page 5 line 100 – page 6 line 110) and explained the rationale for using duocarmycin in discussions (Page 20 line 468 – Page 21 line 475).

2. In Fig. 3A, please complement the panel with image and quantification of pericyte area and avascular/tuft area in the oxygen-challenged retina at P14 (at the point of treatment), to understand the effect of the antibody treatment. Does the antibody treatment arrest pericyte and tuft morphology at their P14 size?

→ Thank you for your delicate comment. In general, oxygen-induced retinopathy mice are exposed to 75% O₂ from P7 to P12. During this time, vascular obliteration is developed in the central retina. And then, mice are returned to normal room air, and the avascular retina becomes relative hypoxic, leading to not only functional vessel regrowth in the retina but also pathological neovascular tuft^{1,2}. The amount of neovascular tuft peaks around P17, while the relatively small amount of neovascular tuft is generated at P14. The avascular area peaks around P12-13, after which it regresses without any treatment due to relative hypoxic stimuli in the OIR mouse model^{3,4}.

(1) *Invest Ophthalmol Vis Sci.* 1994 Jan;35(1):101-111

(2) *Nat Protoc.* 2009;4(11):1565-1573

(3) *J Vis Exp.* 2020 Sep;16;163

(4) *J Vis Exp.* 2021 Apr;2:170

→ Similar tendency in the avascular area and neovascular tuft at P14 (at the point of treatment) and P17 without treatment was observed in our experimental condition of oxygen-induced retinopathy mice. The representative images of whole-mounted retina at P14 and P17 are presented below. It was

found that the NV tuft area was increased and the avascular area was decreased in P17 compared to P14. We provide this data for review, but we are considering not including it in a manuscript.

→ Instead of complement panel with images at P14, we additionally analyzed quantification of pericyte marker (NG2) and added representative images of peripheral retinal vasculature at P17 to evaluate effect of ADC treatment (Fig.3b and 3c).

3. In Fig. 3A, are pericytes undergoing apoptosis after treatment with the bivalent anti-PDGFR β antibody? Please show PDGFR β -positive area in the different treatment conditions, normalized to total vascular area. Examine pericytes both in tufts and in the remaining vascularized area to show whether PDGFR β -positive cells in the tufts are more vulnerable than pericytes associated with the remaining, unaffected vasculature.

→ We appreciate the reviewer's helpful and kind comment. In response to your opinion, we conducted the additional experiment of oxygen-induced retinopathy mice model to evaluate the change of pericytes by quantification of NG2 and PDGFR β coverage with normalization to the total vascular area (IB4+ area). We also compared the result between ADC targeting mPDGFR β and conventional anti-angiogenic agent (1 μ g/eye, bevacizumab, Avastin[®]; Genentech Inc., San Francisco, CA), which is well known as the potent anti-angiogenic effect in OIR mouse model and widely used as an off-label treatment for retinopathy of prematurity.

→ We found that both bevacizumab and ADC targeting mPDGFR β treatment significantly reduced NG2- and PDGFR β positive area at the neovascular tuft compared to vehicle control, while there were

no differences of NG2 and PDGFR β positive area at the peripheral retinal vasculature among three groups. We considered that pericytes in the neovascular tufts are more vulnerable than pericytes in the peripheral retinal vasculature from these data (Supplementary Fig. 2). And this result is likely due to overexpression of PDGFR β of the pericytes in the neovascular tufts than pericytes in the peripheral retinal vasculature, increasing efficiency of ADC targeting mPDGFR β .

4. The authors must have carefully titrated the optimal dose to use for treatment as a higher dose would strip pericytes completely from the retina vasculature, which would cause vascular abnormalities in itself. Please explain and preferably, show the effect of different doses of the bivalent anti-mPDGFR drug-conjugated antibody.

→ We fully agree with the reviewer's opinion. We intravitreally injected PBS or different doses of ADC targeting mPDGFR β (667.7 pg [therapeutic dose in the LI-CNV model] ~ 333.85 ng [500 times higher dose than the therapeutic dose in the LI-CNV model] per eye), and we conducted a blood-retinal barrier leakage test using FITC-dextran and quantified NG2 and PDGFR β coverage to IB4+ vessels, following a single injection.

→ The results showed that the inner blood-retinal barrier in the ADC targeting mPDGFR β treatment group (dose range: 667.7pg ~ 66.77 ng [100 times higher dose than the therapeutic dose in the LI-CNV model] per eye) was intact, and there was no significant difference compared to the vehicle control group, whereas significant fluorescence leakage regions were detected in the ADC targeting mPDGFR β treatment group injecting 500 times higher dose than therapeutic dose in LI-CNV model ($P < 0.01$; Supplementary Fig. 4a and 4b).

→ We also quantified NG2 and PDGFR β coverage to IB4+ vessels to assess pericytes loss. The NG2 and PDGFR β coverage to the retinal vessel was significantly decreased in the ADC targeting mPDGFR β treatment group (333.85 ng/eye) compared with the vehicle control group ($P < 0.01$; Supplementary Fig. 4c and 4d).

→ Considering inner blood-retinal barrier disruption, we require meticulous evaluation to minimize the potential retinal vessel toxicity when determining the optimal dose of ADC for clinical trials.

5. Is inflammation in the retina increased with the antibody treatments?

→ There were no remarkable inflammatory signs (increased inflammatory cells and apoptotic cells, reduced retinal thickness, or retinal folds/damage) in the retinal section images at least our experimental dosage of ADC targeting mPDGFR β (667.7 pg/eye) (Fig. 5a)

Minor

6. *Instead of writing the full name of the bivalent anti-mPDGFRb antibody each time it is mentioned, perhaps the authors could give a short abbreviation.*

→ Thank you for the helpful comment. We shortened the expressions for ‘bispecific anti-mPDGFR β × cotinine scFv-C κ -scFv fusion protein’ and ‘anti-HER2 × cotinine bispecific scFv-C κ -scFv fusion protein’ to ‘anti-mPDGFR β x cotinine’ and ‘anti-HER2 x cotinine’, respectively.

7. *In Fig 1, it is difficult to see the difference in the colors chosen for PDGFRb (pink) and IB4 (red). Please select other colors.*

→ We agree with the reviewer’s suggestion. We changed the color of PDGFR β (Pink to Gray) to make it easier to distinguish PDGFR β and IB4. (Figure 1)

8. *In Fig 1, panel c, in the list of treatments it says mPDGFR-BB which should be mPDGF-BB.*

→ We appreciate your kindness for pointing this out. Accordingly, we corrected the misnomer in Fig. 2, panel c.

Reviewer #3

In this manuscript, the authors investigated whether eradication of pericytes overexpressing mPDGFR β using an ADC could be a promising alternative to conventional anti-VEGF antibody-based therapy for ocular neovascular diseases. The authors prepared and characterized an anti-mPDGFR β ADC equipped with duocarmycin following the method previously reported by them. They subsequently tested the conjugate for the ability to deplete mPDGFR β -expressing MBVP cells in vitro and to suppress retinal neovascularization in mouse models. As the authors concluded, the ADC appears to exhibit mPDGFR β -specific cytotoxicity leading to amelioration of pathologic neovascularization while showing marginal toxicity. ADCs have been developed and used mostly for cancer therapy and ones for non-oncology applications remain to be fully explored. Thus, I believe the manuscript fits the journal's scope and is of great interest to the readership. However, several critical issues regarding ADC preparation and study design were identified. These issues likely dampen or obscure the technical soundness of the work, the clinical potential of the ADC, and the significance of authors' findings. Thus, I suggest the authors to adequately address the concerns listed below.

Major concerns:

1. A conventional anti-VEGF antibody should be tested as a monotherapy control or combination therapy with the ADC in the in vivo treatment studies. Although potential advantages of anti-mPDGFR β agents over anti-VEGF antibodies are described, the manuscript does not fully validate the clinical relevance and potential therapeutic benefits of the ADC due to lack of a comparison with this control group.

→ We fully agree with the reviewer's opinion. In response to the author's suggestion, we newly conducted additional efficacy study treating PBS (vehicle control), ADC targeting mPDGFR β , and anti-angiogenic agent (1 μ g/eye, bevacizumab, Avastin[®]; Genentech Inc., San Francisco, CA), which is well known as potent anti-angiogenic effects¹⁻⁴ without retinal toxicity^{5,6} in both OIR and laser-induced CNV mouse models and widely used as an off-label treatment for retinopathy of prematurity^{7,8} and age-related macular degeneration^{9,10}.

(1) *Clin Exp Ophthalmol*.2019 Jan;47(1):79-87 (Comparison of anti-angiogenic effect among various anti-angiogenic treatments in OIR mouse model)

(2) *Int J Ophthalmol*.2014 Aug 18;7(4):608-613 (OIR mouse model)

(3) *Curr Eye Res*.2015 May;40(6):611-621 (OIR, LI-CNV mouse model)

(4) *Exp Eye Res*.2021 Feb;203:108392. (LI-CNV mouse model)

(5) *Retina*.2008;28(1):46-55 (Bevacizumab, toxicity test, mouse)

(6) *Neurotoxicology*.2008;49(6):1131-1135 (Bevacizumab, toxicity test, mouse)

(7) *Br J Ophthalmol*.2008;92(11):1450-1455 (Retinopathy of prematurity)

(8) *Eur J Ophthalmol*.2019 May;29(3):338-347 (Retinopathy of prematurity)

(9) *N Engl J Med*.2011 May 19;364(20):1897-1908. (Age-related macular degeneration)

(10) *Ophthalmology*.2012 Oct;119(10):2087-2093. (Age-related macular degeneration)

→ In the OIR model, avascular area and neovascular tuft were significantly reduced in the retina injected with bevacizumab and ADC targeting mPDGFR β compared to vehicle control ($P < 0.01$; Supplementary Fig. 2a and 2b).

→ In the laser-induced CNV model, the volume and area of CNV and the volume of NG2- and PDGFR β positive tissue were significantly decreased in mice treated with bevacizumab and ADC targeting mPDGFR β compared with mice administered PBS ($P < 0.01$; Supplementary Fig. 3).

2. Was the ADC pure enough to be used for following in vitro/vivo testing? The authors prepared and characterized the ADC according to the authors' previous report (Ref 46) and apparently did not purify it after complexation with cotinine-duocarmycin. Supplementary Figure 3 of Ref 46 indicates that PRb-CN01 + cot-duo ADC (presumably the same ADC as the one used in this study) contained substantial amount of free cotinine-duocarmycin under physiological conditions. This unpurified payload could contribute to non-specific cell killing as observed for the control ADC in Fig 2c. More importantly, such a mixture may not be appropriate for clinical use. At least the authors should clarify this point and assure in the manuscript that the existence of the free payload does not compromise the study design and conclusion. Otherwise most studies should be performed with a purified ADC.

→ We fully understand and appreciate the reviewer's opinion. Fig 2c represents an *in vitro* experiment in which the drug and the bispecific antibodies are first incubated in 96-well plates for complex formation and subsequently transferred to pre-seeded cells. In such environment, the free payloads will remain exposed to cells, thus allowing random, non-specific toxicity. Therefore, as the reviewer pointed out, the lack of purification of the ADC does present an issue of non-specific toxicity, in *in vitro* environment.

→ For clinical settings, the small size of cotinine-duocarmycin (6.27kDa) is expected to facilitate its fast clearance from the eyes. According to Bakri SJ et al., the half-life of bevacizumab was 4.32 days and ranibizumab, 2.88 days in rabbits with maximum concentrations of 400ug/mL and 162ug/mL at day 1 in the vitreous cavity, respectively. The group states that faster penetration and elimination of ranibizumab from the retina could be attributed to its smaller size (48 kDa) compared to bevacizumab (149 kDa)¹. Therefore, cotinine-duocarmycin, a small molecule with the size of 6.28 kDa, is expected to have even shorter time in the injected area, leaving minimal impact on off-target toxicity. We added these explanations in discussions. (Page 22 line 520– 522)

(1) *Ophthalmology*:2007;114(12):2179–82

→ To clarify, we stated that the lack of purification of ADCs could have contributed to non-specific

killing of cells (Page 22 line 511 - 513)

3. *Fig 2c: although anti-mPDGFR β ADC with or without mPDGF-BB showed enhanced cell killing effect, both control ADC and vehicle also showed concentration-dependent cell killing in 1-10 nM range. Why did these groups, in particular vehicle control, show such unusually high toxicity?*

→ We sincerely apologize for the misleading use of the word ‘vehicle’. ‘Vehicle’ in our *in vitro* study was meant to refer to culture media mixed with cotinine-duocarmycin and free payloads do often demonstrate sub nanomolar IC₅₀ values¹. We have modified the texts and figures to avoid confusion.

(1) *Mol Pharm.* 2015;12(6):1813-1835

4. *The authors state that the in vivo half-lives of the conjugate and its component are 6-15 hours, which are much shorter than those of conventional ADCs (7-14 days). This indicates that frequent and multiple administrations may be required in clinical setting. The authors should discuss whether this shortcoming will negatively impact its practicality in future clinical use.*

→ We thoroughly understand the reviewer’s concern for short half-lives. Many conventional ADCs have targeted whole body systems or cancers with high recurrence, in which case long half-lives and repetitive injection of ADCs would be necessary.

→ However, our ADC proposes selective ablation of pericytes at the site of neovascularization rather than complete ablation of pericytes in the intravitreal area. For such reasons, the short half-life of 6-15 hours of our ADC is believed to be not only sufficient in efficacy, as demonstrated by Fig 3 and Fig 4, but also advantageous in that unwanted entry into the systemic circulation and potential toxicity would be minimized. Furthermore, the effectiveness of our ADC at very small doses may benefit from inhibiting retinal and choroidal neovascularization in future clinical use.

Minor concerns:

1. *Line 104, Page 5: Several ADCs for non-oncology applications have been reported. Such reports should be cited (e.g., Nature 2015, 527:323-8; Bioconjugate Chem. 2018, 29:2357–2369)*

→ We appreciate the reviewer’s insightful opinion and cited relevant works. (Page 5 line 99)

2. *Confocal microscopy: please report how many random images were used for quantification in each study.*

→ We appreciate the reviewer’s helpful comment. We added the description of quantification methods in the case of analyzing randomly chosen images. (Page 10, line 217-222) Furthermore, we added the number of analyzing burn sites in LI-CNV. (Page 11, line 240-241)

3. Line 158, Page 7: please provide cell density of MBVP cells seeded on each well

→ We added the density of MBVP cells for each well. (Page 8 line 161)

4. Line 162, Page 7: please clearly describe what was in “0.006-500 nM” range. Final concentration of each ADC?

→ We appreciate the reviewer’s insightful comment and understand the possibility of confusion. “0.006-500nM” range does refer to the final concentration of each ADC. “0.012nM to 1uM” refers to the concentration of each ADC in 50µL media before incubation with the cells seeded in the same volume of fresh media. We revised the text for clarification. (Page 8 line 161 – 169)

5. Body weight range of the mice at the beginning of each treatment study should be described.

→ We added body weight range of the mice during the toxicity test. (Supplementary Table 2)

6. Line 199, Page 9: were only male mice used as in the other animal studies?

→ Thank you for pointing this out. Sex-related differences in OIR have not been thoroughly explored, possibly because of the difficulty in determining the sex of mouse pups. There have been some reports of no difference in the avascular area and neovascular tuft between male and female pups in the OIR mouse model¹⁻³. Based on these reports, we did not distinguish the sex of newborn pups in the OIR study. We described the sex information of newborn pups. (Page 10, Line 200)

(1) *Invest Ophthalmol Vis Sci.* 1994 Jan;35(1):101-111

(2) *Clin Exp Ophthalmol.* 2001;29(5):323-326

(3) *Sci Rep.* 2017;7:42301

7. Please add to Fig 2 or SI a schematic diagram showing the structures of the ADC and conjugation process. Also please provide the empirically determined drug-to-antibody ratio (DAR) and the method used.

→ We added a schematic diagram of ADC structure and conjugation process in Supplementary Fig.5.

→ The drug-to-antibody ratio (DAR) of the cotinine-duocarmycin used in our study was determined by LC-MS and HPLC analysis. The empirical data and relevant explanations are added as supplementary figure 6 and mentioned in the manuscript (Page 7 line 135-136)

8. *Legend of Fig 2: where is “Scale bars on the bottom left”?*

→ We deeply apologize for the confusion. The reference to scale bars actually belongs to Fig 2b, and we modified the phrase to ‘scale bars in the inset image and on the bottom right’ to avoid confusion.

9. *Line 480, Page 18: Please provide body weight data during this study. I do not see any reason for leaving out this simple information.*

→ We revised description about body weight data. (Page 22, Line 516)

→ We added body weight range during the toxicity test. (Please see Supplementary Table 2)

Reviewers' comments:

Reviewer #1 (Remarks to the Author):

The authors have satisfactorily addressed each of the issues raised in my first review.

Reviewer #2 (Remarks to the Author):

The authors have undertaken a thorough revision which has increased the quality of their study. I have no further criticisms.

Reviewer #3 (Remarks to the Author):

The authors addressed my major concerns with additional data, and this revision significantly increased the importance and impact of the manuscript. One minor concern that remains fully addressed is the determination of the average DAR of the ADC [scFv-Ck-scFv-cot(GSK-duo)4]. The authors determined the DAR of the unconjugated cot(GSK-duo)4 unit to be 4 by reverse-phase HPLC and LC/MS, but it does not necessarily mean the DAR of the ADC was also 4; the DAR should be lower than 4 if the complexation of scFv-Ck-scFv and the cot(GSK-duo)4 unit was not quantitative. I feel this could be the case based on the preparation method they used (1 μ M each, 1:1 mix) and a significant amount of unconjugated cot(GSK-duo)4 observed under physiological conditions in their previous study (Supplementary Fig 3 of Ref 46). Analyzing the ADC under native conditions (e.g., native PAGE, native ESI-MS) could clarify intact scFv (DAR 0)/complex (DAR 4) ratio and eventually allow to calculate the average DAR of the ADC. It is a common practice to determine average DARs in the ADC field if quantitative conjugation is not validated. If the authors think doing so is technically difficult, they should discuss this point and the possibility that the average DAR of the ADC may be lower than 4.

<Point-To-Point responses to reviewers>

Reviewer #3 (Remarks to the Author):

The authors addressed my major concerns with additional data, and this revision significantly increased the importance and impact of the manuscript. One minor concern that remains fully addressed is the determination of the average DAR of the ADC [scFv-Ck-scFv-cot(GSK-duo)4]. The authors determined the DAR of the unconjugated cot(GSK-duo)4 unit to be 4 by reverse-phase HPLC and LC/MS, but it does not necessarily mean the DAR of the ADC was also 4; the DAR should be lower than 4 if the complexation of scFv-Ck-scFv and the cot(GSK-duo)4 unit was not quantitative. I feel this could be the case based on the preparation method they used (1 uM each, 1:1 mix) and a significant amount of unconjugated cot(GSK-duo)4 observed under physiological conditions in their previous study (Supplementary Fig 3 of Ref 46). Analyzing the ADC under native conditions (e.g., native PAGE, native ESI-MS) could clarify intact scFv (DAR 0)/complex (DAR 4) ratio and eventually allow to calculate the average DAR of the ADC. It is a common practice to determine average DARs in the ADC field if quantitative conjugation is not validated. If the authors think doing so is technically difficult, they should discuss this point and the possibility that the average DAR of the ADC may be lower than 4.

→ We understand the reviewer's concern on the average DAR of the ADC. SEC-HPLC was previously used (Supplementary Fig 3 of Ref 46) to quantify antibody-target complexes (in our case, ADC [scFv-Ck-scFv-cot(GSK-duo)4]). To address the concern, we have used another SEC-HPLC to quantify DAR of the ADC. However, we were unable to detect any ADC complex and were only able to detect antibody (scFv-Ck-scFv) and drug (cot(GSK-duo)4) separately. It was reported previously that in some cases, antibody and antigen lose their binding activity and dissociate in the SEC column (1). Hence, it was technically difficult to quantify average DAR of our ADC and hence we have addressed this point in the discussion section (line 475-477).

- (1) Pollastrini, J., Dillon, T.M., Bondarenko, P., Chou, RY. Field flow fractionation for assessing neonatal Fc receptor and Fcγ receptor binding to monoclonal antibodies in solution. *Anal Biochem* 414, 88-98 (2011).

REVIEWERS' COMMENTS:

Reviewer #3 (Remarks to the Author):

The authors have addressed all of my concerns. I believe the manuscript is now ready for acceptance.

<Point-To-Point responses to reviewers>

Reviewer #3 (Remarks to the Author):

The authors addressed my major concerns with additional data, and this revision significantly increased the importance and impact of the manuscript. One minor concern that remains fully addressed is the determination of the average DAR of the ADC [scFv-Ck-scFv-cot(GSK-duo)4]. The authors determined the DAR of the unconjugated cot(GSK-duo)4 unit to be 4 by reverse-phase HPLC and LC/MS, but it does not necessarily mean the DAR of the ADC was also 4; the DAR should be lower than 4 if the complexation of scFv-Ck-scFv and the cot(GSK-duo)4 unit was not quantitative. I feel this could be the case based on the preparation method they used (1 uM each, 1:1 mix) and a significant amount of unconjugated cot(GSK-duo)4 observed under physiological conditions in their previous study (Supplementary Fig 3 of Ref 46). Analyzing the ADC under native conditions (e.g., native PAGE, native ESI-MS) could clarify intact scFv (DAR 0)/complex (DAR 4) ratio and eventually allow to calculate the average DAR of the ADC. It is a common practice to determine average DARs in the ADC field if quantitative conjugation is not validated. If the authors think doing so is technically difficult, they should discuss this point and the possibility that the average DAR of the ADC may be lower than 4.

→ We understand the reviewer's concern on the average DAR of the ADC. SEC-HPLC was previously used (Supplementary Fig 3 of Ref 46) to quantify antibody-target complexes (in our case, ADC [scFv-Ck-scFv-cot(GSK-duo)4]). To address the concern, we have used another SEC-HPLC to quantify DAR of the ADC. However, we were unable to detect any ADC complex and were only able to detect antibody (scFv-Ck-scFv) and drug (cot(GSK-duo)4) separately. It was reported previously that in some cases, antibody and antigen lose their binding activity and dissociate in the SEC column (1). Hence, it was technically difficult to quantify average DAR of our ADC and hence we have addressed this point in the discussion section (line 475-477).

- (1) Pollastrini, J., Dillon, T.M., Bondarenko, P., Chou, RY. Field flow fractionation for assessing neonatal Fc receptor and Fcγ receptor binding to monoclonal antibodies in solution. *Anal Biochem* 414, 88-98 (2011).